# Sample, Scrutinize and Scale:
# Effective Inference-Time Search by Scaling Verification

**Eric Zhao** [1] [2]  **Pranjal Awasthi** [1]  **Sreenivas Gollapudi** [1]

## Abstract

Sampling-based search, a simple paradigm for utilizing test-time compute, involves generating multiple candidate responses and selecting the best one—typically by having models self-verify each response for correctness. In this paper, we study the scaling trends governing sampling-based search. Among our findings is that simply scaling up a minimalist implementation of sampling-based search, using only random sampling and direct self-verification, provides a practical inference method that, for example, elevates the reasoning capabilities of Gemini v1.5 Pro above that of o1-Preview on popular benchmarks. We partially attribute the scalability of sampling-based search to a phenomenon of *implicit scaling*, where sampling a larger pool of responses in turn improves self-verification accuracy. We further identify two useful principles for improving self-verification capabilities with test-time compute: (1) comparing across responses provides helpful signals about the locations of errors and hallucinations, and (2) different model output styles are useful for different contexts—chains of thought are useful for reasoning but harder to verify. We also find that, though accurate verification can be elicited, frontier models demonstrate remarkably weak out-of-box verification capabilities and introduce a benchmark to measure progress on these deficiencies.

## 1. Introduction

Recent advances in language models highlight the importance of test-time compute scaling wherein one uses more compute during inference to enhance reasoning capabilities (OpenAI, 2024; Team, 2025; Agarwal et al., 2024;

Wei et al., 2022; Yao et al., 2023; Akyürek et al., 2024). There are many methods for increasing test-time compute usage, including implicitly encouraging longer responses via reinforcement learning (OpenAI, 2024; Team, 2025) or explicitly via prompting (Wei et al., 2022; Yao et al., 2023). However, *sampling-based search*—an instance of the generate-and-test approach where a model generates many responses in parallel, e.g. via random sampling or delegation, and selects what the model guesses to be the best one—remains one of the most natural and fundamental paradigms. In addition to being complementary with other test-time compute scaling strategies, it also has the unique advantage of being embarrassingly parallel and allowing for arbitrarily scaling: simply sample more responses (Cobbe et al., 2021; Wang et al., 2023).

Though recent works demonstrate the benefits of sampling-based search (Cobbe et al., 2021; Wang et al., 2023; Xue et al., 2023), many questions remain as to what scaling trends govern this fundamental test-time compute scaling strategy. In particular, the practical utility of sampling-based search is limited by *verification*. Whereas prior work have shown that parallel scaling of test-time compute via Pass@k can be remarkably effective (Brown et al., 2024), sampling remains fundamentally bottlenecked by the availability of external verification or the model's ability to self-verify. We are interested in understanding how this bottleneck evolves with scale and how much of the Pass@k - Pass@1 gap can actually be attained in practice.

To develop this understanding, we study a minimalist— yet remarkably effective—instantiation of sampling-based search that uses a language model (Gemini Team, 2024) to both generate a set of candidate responses via random sampling and select the best one by self-verifying each response with natural language. Specifically, we consider the case where models must self-verify their responses to select the best answer, and do not make the strong assumption that

---

[1]Google Research [2]UC Berkeley. Correspondence to: Eric Zhao <eric.zh@berkeley.edu>.

*Proceedings of the 42nd International Conference on Machine Learning*, Vancouver, Canada. PMLR 267, 2025. Copyright 2025 by the author(s).

---

[2]The o1-preview-2024-09-12 numbers in Table 1 use public figures, with MATH and AIME figures sourced from the OpenAI blog post (OpenAI, 2024), and LiveBench figures sourced from the LiveBench leaderboard (livebench.ai). We found o1-Preview performance through the OpenAI API to slightly differ with publicly reported figures, e.g. scoring 26% not 44% on AIME, and scoring 77% not 67% on LiveBench Reasoning.

| Method | AIME | MATH | LiveBench Math | LiveBench Reasoning |
|---|---|---|---|---|
| Pass@1 | 1 / 15 | 426 / 500 | 104 / 200 | 63 / 140 |
| Consistency@200 | 4 / 15 | 460 / 500 | 118 / 200 | 75 / 140 |
| Consistency@1,000 | 3 / 15 | 460 / 500 | 120 / 200 | 73 / 140 |
| Verification@200 | **8** / 15 | **467** / 500 | **135** / 200 | **97** / 140 |
| o1-Preview@1 | 7 / 15 | 428 / 500 | 131 / 200 | 95 / 140 |

*Table 1.* Accuracy rates of the Gemini v1.5 Pro model using sampling-based search (Verification@200) on reasoning benchmarks, compared to other inference methods and o1-Preview performance. Verification@200 consistently improves on Consistency@200 and surpasses o1-Preview.[2] Each score reflects a single run, due to the high expense of search at this scale (see Section 4).

one can access ground-truth answers or symbolic systems that exactly verify correctness. In this setup, we address the question: *what test-time scaling trends emerge as we scale both the number of sampled responses and verification capabilities?* In particular, what are the limits of scaling this simple sampling-based search paradigm and how much does one need to continuously scale verification capability as one scales up search?

**Our findings.** We first identify scaling trends demonstrating that reasoning performance continues to improve with sampling-based search even as test-time compute is scaled well beyond the point where the performance of self-consistency (Wang et al., 2023) saturates. At sufficient scale, even our minimalist implementation provides a significant leap in reasoning accuracy, lifting Gemini v1.5 Pro performance beyond o1-Preview, and Gemini v1.5 Flash beyond Gemini v1.5 Pro, on reasoning benchmarks such as LiveBench (White et al., 2024) and the AIME (MAA, 2024), exhibiting sustained power-law scaling on the latter. This not only highlights the importance of sampling-based search for scaling capability, but also suggests the utility of sampling-based search as a simple baseline on which to compare other test-time compute scaling strategies and measure genuine improvements in models' search capabilities.

We then attribute much of the strong scaling trends of sampling-based search to an *implicit scaling* phenomenon. Contrary to the intuition that sampling more responses should impose a greater burden on the verifier and reduce verification accuracy, we observe that scaling sampling indirectly enhances verification accuracy. At a high-level, this is because well-written responses are easier to verify than poorly written responses, and scaling sampling widens the pool of well-written candidates.

We further identify two effective strategies for scaling verification capabilities using test-time compute: (1) directly comparing candidate responses and (2) task-specific rewriting of candidate responses. The former mitigates a core weakness of language models, which struggle to identify mistakes and hallucinations unless given their locations (Tyen et al., 2024), by leveraging the fact that differences between can-

didate responses provide a strong signal for where errors might be located. The latter leverages our observation of *output style suitability* where chain-of-thought output formats are beneficial when generating responses but harder to verify than more formal, mathematically conventional writing styles. Surprisingly, while effective verification can be easily elicited from frontier models by communicating these strategies, we observe that frontier models have remarkably poor out-of-box verification capabilities and introduce a new benchmark to quantify these deficits.

Historically, self-verification has been believed to be an unreliable means of closing the Pass@k - Pass@1 gap, motivating the use of process-based reward models (Lightman et al., 2024), the reinforcement learning of verifiers (Snell et al., 2024), and other complicated interventions. Our results indicate that self-verification can actually be made quite reliable at scale and in fact is sufficient to extract o1-level performance from non-reasoning models without finetuning, distillation, RL, or custom-trained verifiers.

**Preview and outline.** Table 1 summarizes our first finding: that, with effective self-verification, simply scaling sampling-based search is sufficient to approach state-of-art performance on reasoning and math benchmarks (AIME 2024 (MAA, 2024), LiveBench Math, LiveBench Reasoning (White et al., 2024), and the Berkeley MATH dataset (Hendrycks et al., 2021)). It depicts the accuracy of the Gemini v1.5 Pro model (Gemini Team, 2024) when only one solution[3] is attempted per question (Pass@1), when 200 solutions are attempted and the most common final answer is selected (Consistency@200, (Wang et al., 2023)), and under sampling-based search, when 200 solutions are attempted and scored for correctness with the highest scorer selected (Verification@200, Algorithm 1). With sampling-based search (Verification@200), Gemini v1.5 surpasses the performance of o1-Preview, a model explicitly trained on reasoning problems to leverage significant test-time compute and perform internal search.

---

[3]As we focus on answering reasoning problems, we use "model responses" and "model solutions" interchangeably.

The rest of this paper is devoted to studying the three key factors behind the numbers in Table 1. Section 2.1 analyzes the scalability of sampling-based search, as one varies the compute spent on search and verification; Section 2.2 analyzes the phenomenon of *implicit scaling* that drives this scalability; and Section 3 discusses key principles for scaling self-verification capability. We also highlight deficits in the verification capabilities of frontier models with a new benchmark in Section 5. Further discussion of related work is found in Appendix A.

---

**Algorithm 1** Sampling-Based Search (Verification@$k_{inf}$)

**Input:** Prompt $Q$, model LM, params $k_{inf}, k_{verif}, k_{tie}$.
Populate $\mathcal{S}$ with $k_{inf}$ samples from LM("Answer $Q$").
**for** each candidate response $s_i \in \mathcal{S}$ **do**
    Let $\mathcal{V}_i$ be $k_{verif}$ samples of LM(1[is $s_i$ correct?]).
**end for**
Gather the highest-scored response

$$\mathcal{S}_{Best} = \left\{ s_i \,\middle|\, \mathrm{Avg}(\mathcal{V}_i) \geq \max_{j \in [k_{inf}]} \mathrm{Avg}(\mathcal{V}_j) - 0.05 \right\}.$$

Return $\mathcal{S}_{Best}$ if $|\mathcal{S}_{Best}| = 1$.
**for** each $(s_i, s_j)$ in $\binom{\mathcal{S}_{Best}}{2}$ **do**
    Let $\mathcal{C}_{i,j}$ be $k_{tie}$ samples of LM("Is $s_i$ or $s_j$ correct").
**end for**
Return round-robin winner $s_{i^*}$ of $\{\mathcal{C}_{i,j} \mid s_i, s_j \in \mathcal{S}_{Best}\}$.

---

## 2. Scaling Trends

This section examines how reasoning capability scales with two fundamental test-time compute axes:

- **Search** refers to the compute used to discover candidate solutions. In this section, our knob for scaling search is the number of responses sampled for each reasoning problem ($k_{inf}$ in Algorithm 1).
- **Verification** refers to the compute used to scrutinize candidate solutions. Our knob for scaling verification is the number of verification scores we compute and average over per solution ($k_{verif}$).

For computational reasons, this section uses a streamlined form of Algorithm 1 that omits tie-breaking. This, for example, results in significant underestimates of Verification@k on MATH (see Table 3). All figures are averaged over 20 random seeds, where each run subsamples solutions and verification scores from a primary run that sampled 200 solutions per question and 50 verification scores per solution.

### 2.1. Scaling Trends

Figure 1 provides a heatmap of Verification@k on each benchmark in Table 1 as we scale search and verification.

In addition to clear burn-in costs along both axes of scale, we can observe that the largest performance gains are realized when search and verification are both scaled. These trends also indicate that the performances of sampling-based search, as reported in Table 1, have not yet been scaled to saturation on these benchmarks. This scaling trend is strongest on the AIME benchmark, where performance is bottlenecked by $k$ (search); we attribute this bottleneck to the difficulty of the AIME questions resulting in correct solutions only appearing with very low probability (see Table 2). In particular, we note that of the 7/15 AIME questions where exactly 1-4% of solutions are correct, Verification@200 successfully extracts a correct solution for 5/7 questions.

### 2.2. Implicit Scaling

Scaling sampling-based search along the *search* axis by sampling more solutions, i.e. increasing $k$, should have two effects on performance that partially cancel out: (1) the verifier must discriminate between more solutions, increasing the likelihood of error and (2) the generator is more likely to produce at least one solution that reaches a correct final answer, i.e. Pass@k increases.

To isolate the first effect, we study the model's Verification@k accuracy on "ambiguous" questions: questions where at least one of the model's $k$ candidate solutions reaches the correct final answer (note that Pass@k equals the number of ambiguous questions). Figure 2 and Figure 3 do exactly this, plotting Verification@k accuracy measured only on ambiguous questions from each benchmark. To reduce noise in these figures, we deterministically omit benchmark questions that Consistency@200 answers correctly or where, with high probability, 50 random responses result in either all correct or all incorrect final answers.

After controlling for the growth of Pass@k, we should expect a trend of decreasing accuracy if we increase $k$ but keep the number of verification attempts constant. However, Figure 2 shows the reverse trend: accuracy increases with $k$. This demonstrates an *implicit scaling* of verification accuracy, where increasing the number of generated responses increases not only the chance that at least one response is correct (Pass@k) but also the chance that at least one of the correct responses is of higher quality. Here, quality can be understood as the rigour or flawlessness of a response; a lower quality solution may be generally correct but fail to justify a non-trivial step or make a non-critical error.

Implicit scaling suggests that verification should become more accurate, and sampling-based search should become more effective, with the use of more capable base models that produce more sound reasoning and compelling proofs of correctness. Because the number of ambiguous questions strictly increases with more candidate solutions, the implicit scaling effect also explains the overall accuracy scaling

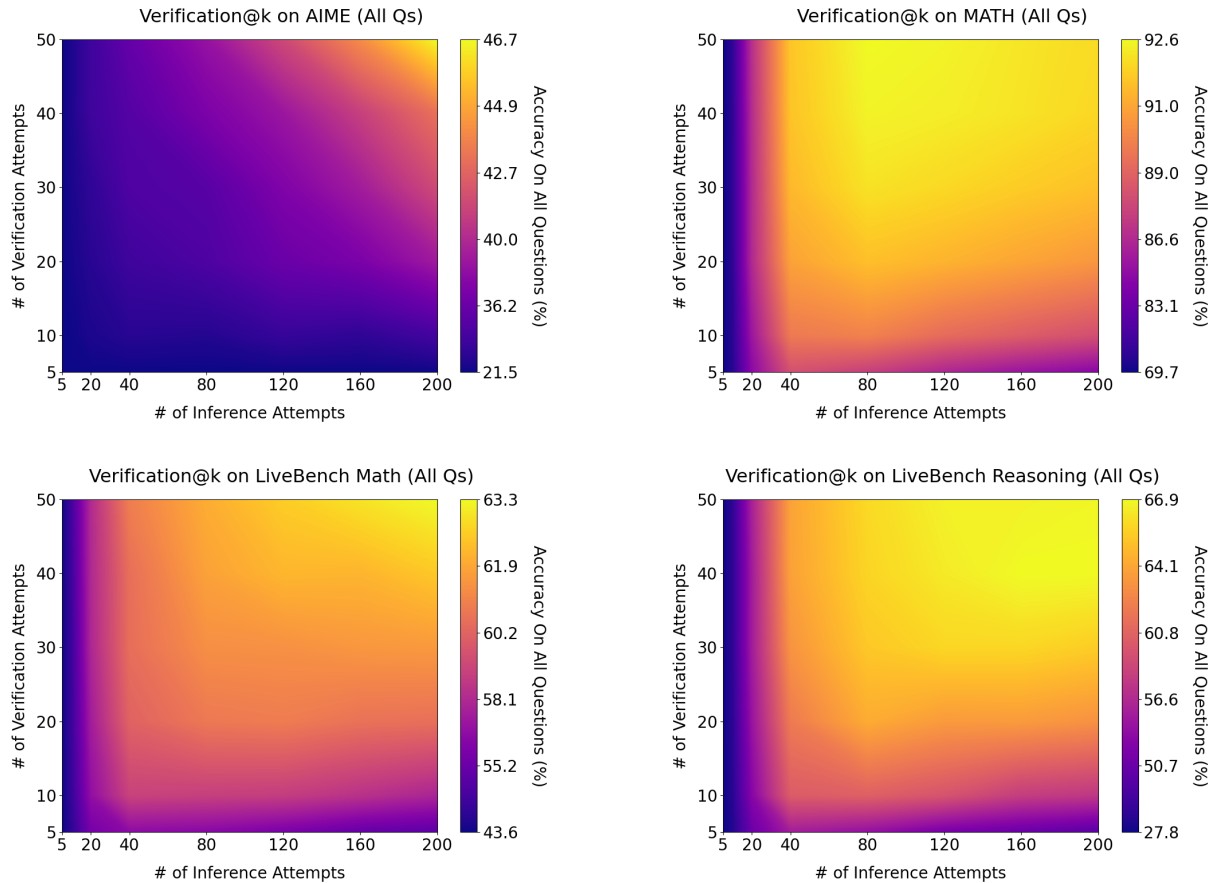

*Figure 1.* Heatmap of Gemini v1.5 Pro accuracy rates using sampling-based search (without tie-breaking) as the number of responses generated (x-axis) and verification attempts (y-axis) increase. Warmer colors indicate higher accuracy (cubic scale). The largest gains occur when scaling both search and verification, with the strongest trend on AIME.

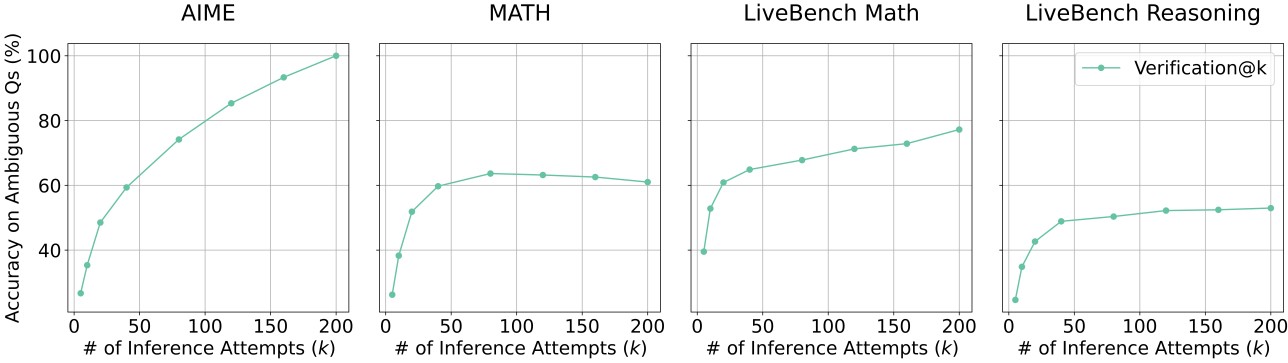

*Figure 2.* Plot of Gemini v1.5 Pro accuracy rates using sampling-based search (without tie-breaking and with $k_{\text{verif}} = 50$) on *ambiguous questions only* as the number of responses generated increases. A question is ambiguous when the model generates at least one candidate response with a correct final answer. Accuracy on ambiguous questions increases with search.

gains in Figure 1: larger $k$ increases both the number of ambiguous questions (Pass@k) and accuracy on the set of ambiguous questions.

## 2.3. The Long Tail of Response Distributions

We can directly observe Verification@k scaling beyond the saturation point of Consistency@k in Figure 4, where we plot their performance after fixing the number of verification attempts at 50. On AIME, the most technically challenging benchmark, Verification@k demonstrates power law scaling even as Consistency@k begins to plateau. The rapid saturation of Consistency@k can be attributed to the fact that, while it is effective at small scales in averaging out noisy mistakes, it necessarily plateaus as it converges on the most probable response; for example, Consistency@50 has the same accuracy as Consistency@10,000 on AIME.

Consider cheaply sampling a vast set of solutions from a weak but ergodic model: Consistency@k is unlikely to return a correct solution, but an effective verifier should still be expected to detect rare but correct solutions in the long-tail of the response distribution. We find an example of this on the AIME 2024 exam, where the Gemini v1.5 model struggles to identify the correct answer to Problem 11 on Exam II. Table 2 shows the final answers from 200 randomly sampled Gemini v1.5 solutions, of which only one is correct ("601," in green).

Consistency returns the incorrect answer of "1" (in red), which appears in over half the responses. In contrast, Verification successfully identifies the solution reaching the correct answer from the response distribution's long-tail, assigning a $\leq 36\%$ score to each solution reaching a final answer of "1" but a 98% score to the single solution reaching "601". Scaling verification capability is key to driving improved search, allowing for discerning between answers that appear correct with 98% vs. 76% confidence. The fact that verification can be used to so effectively leverage the long-tail of model response distributions also suggests that Pass@k, not Pass@1, should be the key performance metric for search applications. Existing post-training techniques (e.g., reinforcement learning from human feedback (RLHF) (Ouyang et al., 2022)) which explicitly optimize for Pass@1 may potentially be doing so at the expense of Pass@k and inhibiting search capability.

## 3. Effective Self-Verification

In the process of scaling sampling-based search, we identified two general principles for eliciting more accurate self-verification, that may be of independent interest.

1. *Compare responses to localize errors.* Disagreements between candidate solutions strongly signal the potential locations of their errors. This can be leveraged to combat the fact that language models have low recall (i.e., often overlook) when asked to identify mistakes and hallucinations (Tyen et al., 2024; Kamoi et al., 2024a), as models are able to identify errors when provided their locations (Tyen et al., 2024). We can improve the self-verification of a candidate response by providing the verifier with other responses to compare the candidate against—an instance of implicit scaling.

2. *Rewrite responses for output style suitability.* The optimal output style of a language model should depend on the task. Writing in a linear chain of thought— which includes detailing reasoning before committing to a claim—is effective when generating responses (search) (Wei et al., 2022). However, responses are easier to verify when written rigorously, hierarchically, and modularly. This can be leveraged by having verifiers first rewrite candidate responses in, e.g., an expanded mathematically conventional theorem-lemma-proof format rather than directly evaluating chains-of-thought.

These principles also provide levers for scaling self-verification capability with test-time compute, including by (1) sampling and providing verifiers with more responses to compare between and (2) rewriting responses with increasing rigour and structure.

### 3.1. Sampling-Based Search Implementation

We now detail our minimalist implementation of sampling-based search (summarized in Algorithm 1) that uses only parallelizable blackbox queries to a language model. It generates candidate responses by randomly sampling from models and select responses by asking models to self-verify; prompts are identical across all benchmarks.

**Step 1: Generate Candidate Responses.** A language model generates $k_{\text{inf}}$ candidate responses (candidate solutions) in parallel to each question, using temperature $\sigma_{\text{inf}}$.

**Step 2: Verify Candidate Responses.** A language model generates $k_{\text{verif}}$ binary "verification scores" for each candidate in parallel, indicating whether its final answer is correct. Each scoring attempt is a single conversation thread that rewrites the response as a theorem, supporting lemmas, and proofs (examples in Appendix F) and systematically scans for errors. The highest scoring response is selected.

**Tie-Break: Compare Candidate Responses.** When the three highest scoring candidates score within 5% of one another and disagree on the final answer, a language model directly compares the responses in pairwise matchups. Each matchup is a single conversation thread that identifies where responses diverge and, at each such point, determines which side is correct. Each matchup is repeated $k_{\text{tie}} = 100$ times, and the response with the most wins is selected.

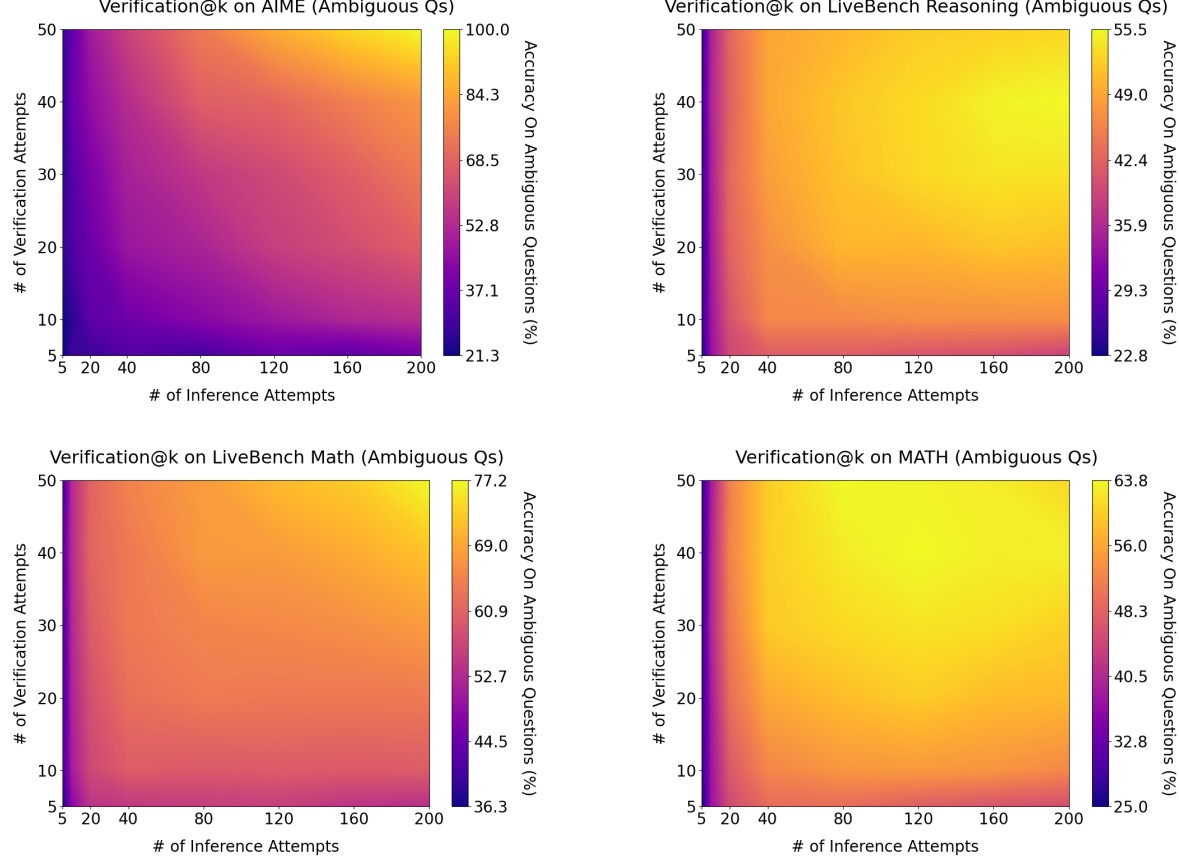

*Figure 3.* Heatmap of Gemini v1.5 Pro accuracy rates using sampling-based search (without tie-breaking) on *ambiguous questions only* as the number of responses generated (x-axis) and verification attempts (y-axis) increase. Warmer colors indicate higher accuracy (linear scale). A question is ambiguous when the model generates at least one candidate response with a correct final answer. Accuracy on ambiguous questions increases with search (x-axis).

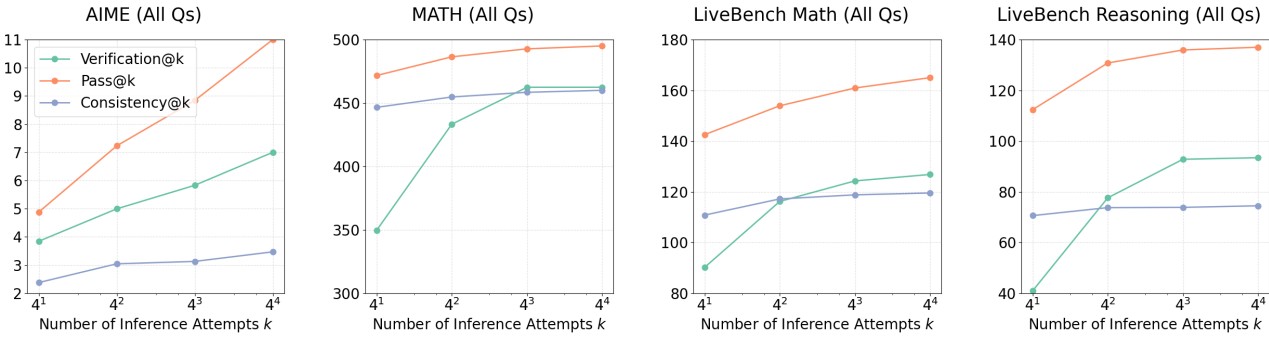

*Figure 4.* Line graph depicting the accuracy rates of the Gemini v1.5 Pro model using sampling-based search as the number of candidate responses generated is scaled upwards. The number of verification attempts is fixed at 50 for all plots. The depicted accuracies are obtained without tie-breaking and may be lower than reported elsewhere. Verification@k improves with $k$ even when Consistency@k stagnates on AIME and LiveBench Reasoning.

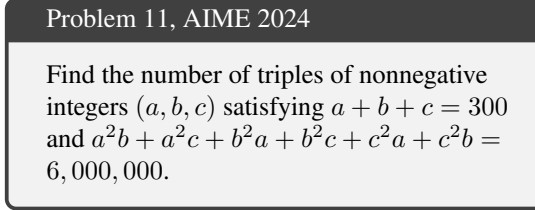

**Problem 11, AIME 2024**

Find the number of triples of nonnegative integers $(a, b, c)$ satisfying $a + b + c = 300$ and $a^2b + a^2c + b^2a + b^2c + c^2a + c^2b = 6,000,000$.

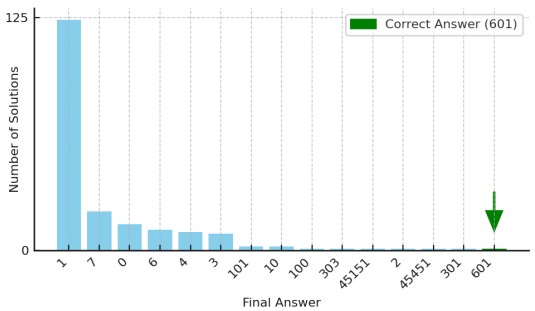

| Verification Score | Final Answer | | # Solutions |
|---|---|---|---|
| 0.98 | 601 | (Correct) | 1 |
| 0.76 | 6 | (Wrong) | 11 |
| 0.52 | 0 | (Wrong) | 14 |
| 0.40 | 7 | (Wrong) | 21 |
| 0.38 | 4 | (Wrong) | 10 |
| 0.36 | 1 | (Wrong) | 124 |
| 0.22 | 10 | (Wrong) | 2 |
| 0.20 | 3 | (Wrong) | 9 |
| 0.18 | 301 | (Wrong) | 1 |
| 0.16 | 45451 | (Wrong) | 1 |
| 0.14 | 101 | (Wrong) | 2 |
| 0.06 | 2 | (Wrong) | 1 |
| 0.04 | 45151 | (Wrong) | 1 |
| 0.04 | 303 | (Wrong) | 1 |
| 0.00 | 100 | (Wrong) | 1 |

*Table 2.* The final answers identified by the Gemini v1.5 Pro model to Problem 11 on AIME 2024, sorted by verification score and annotated with their multiplicity in 200 solution generations. The correct final answer (green) is only found by 1 generated response whereas Consistency@200 selects an incorrect final answer (red) that is found by 124 generated responses.

### 3.2. Ablation Studies

We can individually ablate the practices of comparing and rewriting candidate responses to confirm their role in eliciting greater verification capability.

**Ablating comparisons.** The step of asking models to directly compare candidate solutions with similar verification scores significantly increases sampling-based search performance. This is demonstrated in Table 3, where we depict the accuracy rates from Table 1 alongside the accuracy rates after ablating the tie-breaking step. These comparisons have the greatest impact when models struggle from low recall and excessively assign high verification scores. On the MATH benchmark, which sees the greatest lift from comparisons, the average verification score of the top 3 candidate responses is nearly 90%.

**Ablating rewritings.** We explored a limited number of prompts for self-verification, including prompts which omit instructing the model to rewrite responses. We did not perform further prompt optimization and expect refinements would boost accuracy. Table 4 shows each prompt's probability of mislabeling correct solutions (false positive) and incorrect solutions (false negative), with the former generally having a more severe impact on downstream performance. We evaluated these prompts on 1,080 candidate responses to 54 level-5 questions from the MATH training split, and 120 candidate responses to 6 questions from AIME 2023. A response is marked as incorrect if, of 20 verification attempts, the number finding an error in the solution exceeds the equal error rate threshold.

*Main* refers to manually written prompts used in our experiments. *Shortened* refers to a shorter variant of "Main" that omits, e.g., instructions to avoid truncation. *Without Rewrite* refers to a variant of "Main" that omits instructing the verifier to first rewrite responses. *Split-Context* refers to a variant of "Main" that creates separate conversation threads to individually verify pieces of the response.

The gap between the performance of "Main" and "Without Rewrite" demonstrates that ablating the rewriting of solutions negatively impacts verification performance. Similarly, the gap with "Split-Context" demonstrates that splitting the verification process into separate conversation threads sharply decreases performance due to low precision, which we attribute to miscalibration.

### 4. Technical Details

All experiments are run on Google Cloud with Gemini v1.5-Pro-002 and Gemini v1.5-Flash-002 models dated to September 2024. Unless otherwise specified, the default parameters for our implementation of sampling-based search (Section 3) are $k_{\text{inf}} = 200$, $\sigma_{\text{inf}} = 1.5$, $k_{\text{verif}} = 50$, $\sigma_{\text{verif}} = 1$, and a maximum of 8,192 output tokens per query. For all benchmarks, the scoring of candidate responses is performed using a language model rather than literal string comparison; details are in Appendix C.2.

**Preliminary scoring.** When generating $k_{\text{verif}} = 50$ verification scores per candidate solution is too expensive, we first generate $k_{\text{verif}} = 10$ preliminary verification scores and discard candidate solutions with an average score below

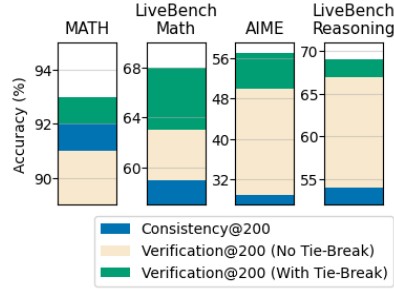

| Dataset | Cons@200 | Verification@200 | |
| --- | --- | --- | --- |
| | | Without | With Tie-Break |
| MATH | 460 / 500 | 457 / 500 | 467 / 500 |
| LiveBench Math | 118 / 200 | 125 / 200 | 135 / 200 |
| LiveBench Reasoning | 75 / 140 | 94 / 140 | 97 / 140 |
| AIME | 4 / 14 | 7 / 14 | 8 / 14 |

*Table 3.* Accuracy rates of Gemini v1.5 Pro using sampling-based search, with and without tie-breaking. Tie-breaking provides most of Verification@200's gains on Consistency@200 (Cons@200) on MATH and LiveBench Math, and smaller gains on AIME and LiveBench Reasoning.

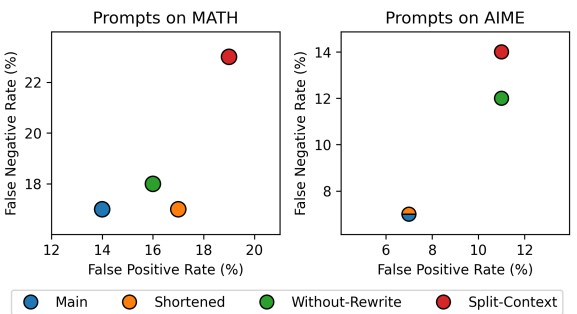

| Prompt Style | MATH | | AIME | |
| --- | --- | --- | --- | --- |
| | FPR | FNR | FPR | FNR |
| Main | 14% | 17% | 7% | 7% |
| Shortened | 17% | 17% | 7% | 7% |
| Without Rewrite | 16% | 18% | 11% | 12% |
| Split-Context | 19% | 23% | 11% | 14% |

*Table 4.* Verification scoring accuracy rates of the Gemini v1.5 Pro model for various prompts. False positive rate (FPR) refers to how often a correct response is labeled as incorrect; false negative rate (FNR) refers to how often an incorrect response is labeled as correct.

0.2. If a final answer is represented by more than 15 candidate responses, only the top 15–as measured by average preliminary score, tie-breaking randomly–are kept. This results in a smaller pool of candidate solutions for which we compute all $k_{\text{verif}} = 50$ verification scores. Preliminary scoring is used on all datasets except AIME.

**Compute.** On AIME, the verification process involves 32,000 characters (roughly 13,000 tokens) of model output. Extrapolating from these figures, running the full sampling-based search pipeline on a question for $k_{\text{inf}} = 200$ and $k_{\text{verif}} = 50$ requires $200 \cdot 50 \cdot 13,000 \approx 130$M output tokens. At around $5/1M output tokens (public pricing of Gemini v1.5 Pro), this is approximately $650 in cost. Preliminary scoring reduce output token usage by 70%, with a per-question cost of $200. The use of Gemini Flash for verification decreases cost to $12 per question.

**Datasets.** Our MATH benchmark consists of 500 questions from the PRM800K (Lightman et al., 2024) test split of MATH (Hendrycks et al., 2021). Our LiveBench Math benchmark consists of 200 random questions from the 368 available as of October 21st 2024, including AMC12 2023, AIME 2024, SMC 2023, USAMO 2023, IMO 2023, and synthetic math questions (White et al., 2024). Our LiveBench Reasoning benchmark consists of 140 questions

from the 150 available as of October 21st 2024, including Zebra puzzles, Web-Of-Lies, and Spatial reasoning (White et al., 2024). Our AIME benchmark consists of the 15 questions in the 2024 Exam II (MAA, 2024).

## 5. A Verification Benchmark

Frontier language models demonstrate a remarkable mismatch between their problem-solving capabilities and poor out-of-box verification capabilities. These limitations have largely been attributed to the inability of current language models to self-diagnose hallucinations or enforce rigour (Zhang et al., 2023; Orgad et al., 2024; Snyder et al., 2024; Kamoi et al., 2024a; Tyen et al., 2024; Huang et al., 2024). However, our findings that models can be directed to accurately perform verifications at scale suggest that these out-of-box limitations can be addressed with standard methods like instruction tuning. We compiled a set of challenging reasoning problems and candidate solutions to provide a benchmark for these deficits.

Each entry in this benchmark consists of a question, a correct candidate response, and an incorrect candidate response, and is manually curated from the residuals of our sampling-based search experiments (Section 3). Example entries from this benchmark can be found in Appendix E.

| Model | Metric | Scoring Accuracy | | | Comparison Accuracy |
|---|---|---|---|---|---|
| | | Correct | Wrong | Flawed | |
| GPT-4o | Pass@1 | 76.5% | 31.0% | 22.2% | 43.2% |
| | Consistency@5 | 77.4% | 30.0% | 11.1% | 35.4% |
| Claude 3.5 Sonnet | Pass@1 | 89.6% | 22.5% | 33.3% | 56.1% |
| | Consistency@5 | 90.3% | 17.5% | 33.3% | 61.2% |
| o1-preview | Pass@1 | 100% | 68.8% | 80.0% | 84.5% |
| | Consistency@5 | 100% | 79.4% | 88.8% | 92% |
| Gemini 2.0 Flash | Pass@1 | 73.5% | 44.5% | 60% | 58% |
| | Consistency@5 | 77.4% | 42.5% | 66.6% | 58.7% |
| Gemini 2.0 Thinking Flash | Pass@1 | 75.4% | 56.5% | 53.3% | 80% |
| | Consistency@5 | 77.4% | 55% | 55.5% | 89.1% |
| Random guessing | | 80% | 20% | 20% | 50% |

*Table 5.* Accuracy rates of commercial language models on our verification benchmark. For the task of response scoring (Scoring Accuracy), accuracy rates are broken down for entries that require identifying a correct response as being correct (Correct), entries that require identifying a wrong response as being wrong (Wrong), and entries that require identifying a wrong response that coincidentally reaches the correct answer as being wrong (Flawed). GPT-4o and Claude 3.5 Sonnet only perform marginally better than random guessing across all tasks. o1-Preview performs better, but still fails to identify 20-30% of wrong responses.

Our benchmark studies verification accuracy on two tasks:

1. **Scoring task.** When given only the question and one of the responses, is the model able to discern the correctness of the response?
2. **Comparison task.** When provided the whole tuple with the correctness labels of the responses masked and a guarantee that at least one response is correct, is the model able to discern which response is correct and which is incorrect?

The scoring task is also evaluated over a separate set of (question, response) pairs where the response reaches the correct final answer by coincidence but contains fatal errors and should be labeled by a reasonable verifier as being incorrect; an example can be found in Appendix E. In the scoring task, models are provided only with the task description; in the comparison task, models are provided only with the task description and a suggestion to identify disagreements between responses in its reasoning.

Table 5 lists the baseline performances of current commercial model offerings on this benchmark. Gemini v1.5 Pro is omitted from the benchmark as the entries in the benchmark are curated from the residuals of Gemini v1.5 Pro. The prompts used in Table 5 are provided in Appendix C.4.

As we previously observed, and has been noted in prior works (Tyen et al., 2024; Kamoi et al., 2024a), verification errors are typically due to low recall. Even the easier comparison task, models perform only marginally better—and often worse—than random chance. In many cases, Consistency@5 performs worse than one-shot inference be-

cause Consistency simply averages out noise from an output distribution, meaning that a model biased towards producing an incorrect answer will do so with higher probability under Consistency. Addressing these deficits in verification capabilities—which we see as low-hanging fruit for post-training—would enable not only better sampling-based search, but also other downstream applications of verification including reinforcement learning (e.g. OpenAI, 2024; Team, 2025), data flywheeling (e.g., Welleck et al., 2022), and end-user experience (further discussion in Section A).

## 6. Conclusion

This paper studied the scaling trends governing sampling-based search with self-verification, finding that (1) it scales remarkably well even with simple implementations, (2) *implicit scaling* plays a big role in this scalability, and (3) self-verification capability can be scaled with test-time compute using two key principles: comparisons localize errors, and responses should be rewritten for output style suitability. To this end, we scaled a minimalist, embarrassingly parallel implementation of sampling-based search that, with sufficient test-time compute, is sufficient to attain state-of-art performance on a range of reasoning benchmarks. Given that it complements other test-time compute scaling strategies, is parallelizable and allows for arbitrarily scaling, and admits simple implementations that are demonstrably effective, sampling-based search will play a crucial role as language models are tasked with solving increasingly complex problems with increasingly large compute budgets.

## Acknowledgments

This material is based on work supported by the National Science Foundation Graduate Research Fellowship Program under Grant No. DGE 2146752 and by a Google PhD Fellowship (EZ).

## Impact Statement

This paper presents work whose goal is to advance the field of Machine Learning. There are many potential societal consequences of our work, none which we feel must be specifically highlighted here.

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

# A. Related Work

**Test-time compute.**   Many of the recent advances in language model reasoning capabilities can be traced to increasing use of *test-time compute*. Inference strategies like chain-of-thought reasoning (Wei et al., 2022), tree-of-thoughts (Yao et al., 2023) and self-critique (Valmeekam et al., 2023) result in improved reasoning performance at the cost of forming longer responses. Reinforcement learning has emerged as a particularly successful strategy for effectively leveraging more test-time compute, wherein models learn from exploration to form lengthy chain-of-thought outputs that incorporate backtracking and search, despite not being explicitly taught to do so (OpenAI, 2024; Team, 2025). Inference-time model adaptation, whether through many-shot learning (Agarwal et al., 2024; Anil et al., 2024) or finetuning (Akyürek et al., 2024), provides another avenue when training data is available. We study sampling-based search: obtain a set of candidate responses from a model and apply an aggregation method to select a response, such as self-consistency/plurality voting (Wang et al., 2023) or selecting a response with a reward/verifier model (Cobbe et al., 2021). These various methods for scaling test-time compute are complementary; for example, sampling-based search can also be used on models trained to produce longer outputs. We note that it is possible for models trained to produce long chains of thought to perform something resembling sampling-based search internally, in which case we still expect our observed scaling trends to hold. However, we also expect explicit sampling-based search will remain indispensable, due to its greater parallelism and robustness than internally implemented search.

**Scaling sampling-based search.**   The paradigm of sampling-based search provides three main knobs for scaling: generation, sampling, and selection. While the cost of *generating* each individual response can be scaled with previously mentioned interventions, such as chain-of-thought (e.g. Wei et al., 2022), reinforcement learning (e.g. OpenAI, 2024), or inference-time adaptation (e.g. Anil et al., 2024), the cost of *sampling* a set of responses can be scaled by increasing the number of responses generated (Wang et al., 2023; Snell et al., 2024; Brown et al., 2024). We use random sampling to generate each set of candidate responses, which means the latter corresponds to simply taking more random draws. Whereas prior work on parallel test-time compute scaling via Pass@k, such as Brown et al. (2024), has focused on understanding how Pass@k scales, we are concerned with how much of the Pass@k - Pass@1 gap can actually be attained in practice. However, this sampling can also be implemented in an agentic fashion, with a central model delegating the generation of responses so as to perform search more systematically. The process of *selecting* a response can be scaled by using more expensive rules: self-consistency provides a simple plurality voting rule at the lowest-cost end of the spectrum (Wang et al., 2023), while language model self-verification (e.g. Xue et al., 2023, see below) and learned verification/reward models (e.g. Cobbe et al., 2021, see below) provide a range of selection strategies that vary in cost and capability. For more fine-grained control over the scaling of self-verification in our experiments, we apply plurality voting (Wang et al., 2023) to self-verification and vary our number of verification attempts per response.

**Verification of language model outputs.**   A large body of recent work has studied the self-verification capabilities of large language models (e.g., Cobbe et al., 2021; Kadavath et al., 2022; Saunders et al., 2022; Kim et al., 2023; Xie et al., 2023; Weng et al., 2023; Zhang et al., 2023; Xue et al., 2023; Li et al., 2023; Liu et al., 2024; Chow et al., 2024; Jiang et al., 2024; Dhuliawala et al., 2024; Snyder et al., 2024; Wu et al., 2024b; Huang et al., 2024; Kamoi et al., 2024a;b; Orgad et al., 2024; Wen et al., 2024; Tyen et al., 2024; Chen et al., 2024; Kumar et al., 2024; Qu et al., 2024; Zhang et al., 2024; Ko et al., 2025; Havrilla et al., 2024). While some works—including ours—simply ask models to perform verification and parse the response, others have proposed custom methods of performing self-verification, including: recreating the problem from the response (Xue et al., 2023; Wu et al., 2024b), masking and re-filling parts of the response (Weng et al., 2023; Jiang et al., 2024), creating a rubric (Dhuliawala et al., 2024), or asking models to choose from options (Xie et al., 2023; Chen et al., 2024). Our work does not focus on optimizing for self-verification or advocate for any particular strategy. However, in the course of performing our scaling study, we did identify several previously unstudied principles of self-verification that only arise at sufficiently large scale and may be of independent interest, including implicit scaling, output style suitability, and the importance of directly comparing responses. Other related bodies of work study the learning of verifiers, often on top of a pretrained large language model (e.g. Cobbe et al., 2021; Saunders et al., 2022; Li et al., 2023; Havrilla et al., 2024; Kumar et al., 2024; Qu et al., 2024; Chow et al., 2024; Zhang et al., 2024), and the use of external tools for verification (e.g. Min et al., 2023; Gou et al., 2024; Gao et al., 2024; Kim et al., 2023). We did not train customized verification models or permit verifier use of external tools in the listed experiments, as we found blackbox model access to be sufficient for effective verification at scale. The limitations of model self-verification capabilities are also well-studied (Kamoi et al., 2024a; Tyen et al., 2024; Huang et al., 2024; Wu et al., 2024a), and can be remedied with external information (Huang et al., 2024) or hints for localizing errors (Tyen et al., 2024). Models especially struggle with self-diagnosing hallucinations (Zhang et al.,

2023; Orgad et al., 2024; Snyder et al., 2024), despite awareness of their own limitations (Kadavath et al., 2022), and are often incentivized to obfuscate errors (Wen et al., 2024).

**Applications of verification.** In addition to being used to select from candidate responses (Cobbe et al., 2021; Li et al., 2023; Weng et al., 2023; Jiang et al., 2024; Chen et al., 2024; Xie et al., 2023), verifiers can be used to guide iterative improvements to a model's output by providing feedback to the generating model (Kim et al., 2023; Xue et al., 2023; Valmeekam et al., 2023; Wu et al., 2024b; Huang et al., 2024; Dhuliawala et al., 2024; Stechly et al., 2024a;b; Qu et al., 2024; Havrilla et al., 2024; Ko et al., 2025). Another important application of verification is in enhancing model capabilities. For example, verification results for model outputs can be fed back into models as feedback via in-context reinforcement learning (Shinn et al., 2023), reinforcement learning (Uesato et al., 2022; Peng et al., 2023; Madaan et al., 2023; Kumar et al., 2024; Chow et al., 2024), or finetuning (Welleck et al., 2022; Paul et al., 2024; An et al., 2024; Singh et al., 2024), in an approach known as data flywheeling. Verification has also been explored as a means of encouraging models to produce better written responses (Anil et al., 2021; Kirchner et al., 2024). From a product perspective, verification capabilities are also important to the workflow of end users (Collins et al.).

## B. Additional Experiments

### B.1. Smaller Models

| Model | Method | AIME | MATH | LiveBench Math | LiveBench Reasoning |
|-------|--------|------|------|----------------|---------------------|
| Pro v1.5 | Pass@1 | 1 / 15 | 426 / 500 | 104 / 200 | 63 / 140 |
| | Consistency@200 | 4 / 15 | 460 / 500 | 118 / 200 | 75 / 140 |
| | Verification@200 | 8 / 15 | 467 / 500 | 135 / 200 | 97 / 140 |
| Flash v1.5 | Pass@1 | 2 / 15 | 407 / 500 | 96 / 200 | 65 / 140 |
| | Consistency@200 | 3 / 15 | 440 / 500 | 92 / 200 | 84 / 140 |
| | Verification@200 | 5 / 15 | 445 / 500 | 104 / 200 | 84 / 140 |
| Pro+Flash v1.5 | Verification@200 | 7 / 15 | 456 / 500 | 119 / 200 | 84 / 140 |

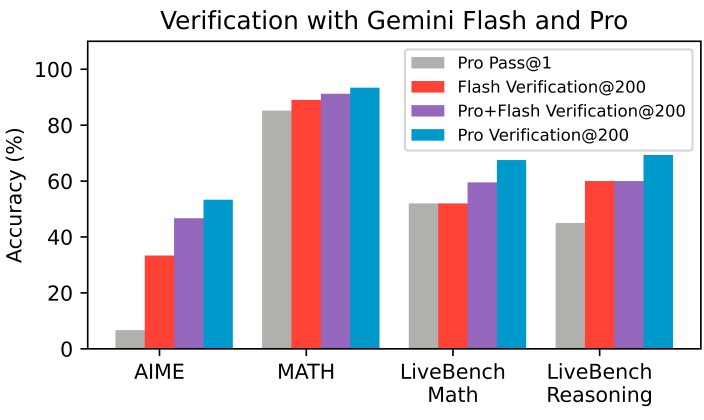
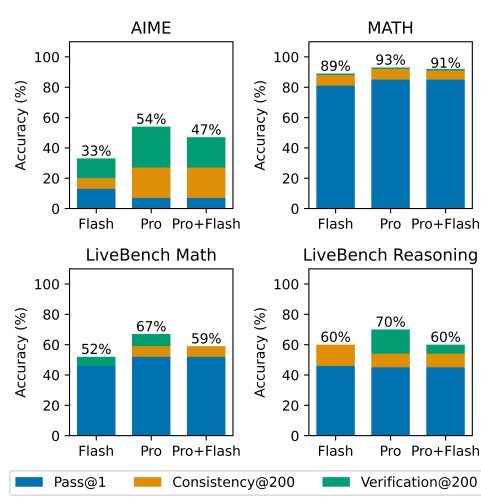

*Table 6.* Accuracy rates with sampling-based search using either the Gemini v1.5 Pro model to both generate and verify responses (Pro), Gemini v1.5 Flash to both generate and verify responses (Flash), or Gemini v1.5 Pro model to generate responses and v1.5 Flash to verify responses (Pro+Flash). Verification@200 exceeds Consistency@200 for all model choices, while Pro+Flash Verification@200 matches or exceeds Pro Consistency@200.

We also observe sampling-based search to be a powerful tool for enhancing smaller, lower-cost models. Here, we apply sampling-based search to Gemini v1.5 Flash model, which has a nearly 20x lower inference cost than Gemini v1.5 Pro. Table 6 lists the performance of using the Flash model to evaluate candidate responses generated by the Pro model (Pro+Flash), and the performance of using the Flash model end-to-end for sampling-based search (Flash). Sampling-based

search still provides a significant improvement in performance for both Flash and Pro+Flash. Moreover, Verification@200 still provides significant improvements over Consistency@200, albeit lesser in magnitude than for end-to-end use of Gemini Pro. In addition, Flash Verification@200 using Gemini Flash is competitive with Pro Consistency@200, while Pro+Flash Verification@200 exceeds Pro Consistency@200. We highlight that Pro+Flash Verification@200 has roughly the compute cost of Consistency@500—as our sampling-based search implementation is minimally optimized for efficiency, we expect costs to further decrease.

## B.2. Performance by Subtask

| Dataset | Cons@200 | Verif@200 | Improvement (%) | | Pass@200 | # Questions |
|---|---|---|---|---|---|---|
| | | | (Abs) | (Rel) | | |
| Berkeley MATH | 92.0% | 93.4% | 2% ↑ | 20.0% | 99.0% | 500 |
| AIME 2024 | 26.7% | 53.3% | 100% ↑ | 57.1% | 73.3% | 15 |
| Web-of-Lies-v2* | 75.5% | 91.8% | 22% ↑ | 66.5% | 100.0% | 49 |
| Spatial* | 33.3% | 46.7% | 40% ↑ | 21.5% | 95.6% | 45 |
| Zebra Puzzle* | 50.0% | 67.4% | 35% ↑ | 36.4% | 97.8% | 46 |
| Competition[†] | 66.2% | 83.1% | 26% ↑ | 63.1% | 93.0% | 71 |
| AMPS Hard[†] | 70.6% | 77.7% | 10% ↑ | 33.5% | 91.8% | 85 |
| Olympiad[†] | 25.0% | 22.7% | 9% ↓ | -11.2% | 45.5% | 44 |

*Table 7.* The Pass@200, Consistency@200 (Cons@200), and Verification@200 (Verif@200) accuracy rates of the Gemini v1.5 Pro model using sampling-based search. LiveBench Math[†] and LiveBench Reasoning[*] numbers are divided per task. Absolute % Increase (Abs) is the percentage improvement of Verification@200 over Consistency@200. Relative % Increase (Rel) is (Verification@200 - Consistency@200) / (Pass@200 - Consistency@200).

The LiveBench benchmarks each consist of multiple subtasks. In Table 7, we break down the numbers reported in Table 1 for each of these subtasks. We also provide in Table 7 the Pass@200 scores of the Gemini Pro model, which measure the probability that of 200 attempted responses to a question at least one is correct. Pass@200 upper bounds what one can hope to achieve through Verification or Consistency. Verification provides the greatest gains on AIME 2024, Web-of-Lies, Competition, and Zebra Puzzle. In contrast, Verification does not improve on Consistency on the Olympiad task of the LiveBench Math benchmark. We attribute this to the unique question design of LiveBench Olympiad task questions, which is incompatible with our implementation of Verification (see Appendix B.4).

## B.3. Temperature Tuning

After attempting temperature tuning on the training split of the MATH dataset, we found that the choices of temperatures $\sigma_{\text{inf}}$ and $\sigma_{\text{verif}}$ did not significantly affect performance. In Table 8 and Table 9, we compare the post-verification accuracy of our pipeline for various temperature choices. The Verification@20 figures are obtained without running tie-breaking and with only 20 verification attempts.

| Inference Temp $\sigma_{\text{inf}}$ | Pass@20 | Consistency@20 | Verification@20 |
|---|---|---|---|
| 0.2 | 89/100 | 76/100 | 82/100 |
| 1.0 | 94/100 | 73/100 | 80/100 |
| 1.5 | 89/100 | 73/100 | 79/100 |
| 2.0 | 89/100 | 75/100 | 82/100 |

*Table 8.* Accuracy rates of the Gemini v1.5 Pro model on the training split of MATH using different methods of selecting from 20 generated responses and varying the temperature used to generate the responses. The temperature used for verification attempts is fixed at $\sigma_{\text{verif}} = 1.0$. Inference temperature does not significantly affect downstream performance.

| Verification Temp $\sigma_{\text{verif}}$ | Pass@20 | Consistency@20 | Verification@20 |
|---|---|---|---|
| 0.2 | 89/100 | 73/100 | 76/100 |
| 1.0 | 89/100 | 73/100 | 79/100 |
| 2.0 | 89/100 | 73/100 | 79/100 |

*Table 9.* Accuracy rates of the Gemini v1.5 Pro model on the training split of MATH using different methods of selecting from 20 generated responses and varying the temperature used to for verification attempts. The temperature used for generating responses is fixed at $\sigma_{\text{inf}} = 1.5$. Verification temperature does not significantly affect downstream performance.

### B.4. Olympiad LiveBench Math Subtask

The one task for which we saw no lift from verification is the Olympiad questions from LiveBench MATH. These questions are not formatted as open-ended problems. Rather, they take a very specific form of asking one to fill in a pre-written proof from a menu of expression options, and to output a specific sequence of indices corresponding to these options. This is incompatible with our verification pipeline, which asks the verification model to rewrite candidate responses in a theorem-lemma format where the theorem states the final answer. For example, the final answer to the Olympiad question at the bottom of this section is the following sequence:

19,32,20,2,14,1,27,21,31,36,3,30,5,16,29,34,7,4,6,18,15,22,9,25,28,35,26,8,13,24,23,17,33,11,10,12 .

---

**Representative example of Olympiad task from LiveBench MATH**

You are given a question and its solution. The solution however has its formulae masked out using the tag ¡missing X¿ where X indicates the identifier for the missing tag. You are also given a list of formulae in latex in the format "¡expression Y¿ = latex code" where Y is the identifier for the formula. Your task is to match the formulae to the missing tags in the solution. Think step by step out loud as to what the answer should be. If you are not sure, give your best guess. Your answer should be in the form of a list of numbers, e.g., 5, 22, 3, ..., corresponding to the expression identifiers that fill the missing parts. For example, if your answer starts as 5, 22, 3, ..., then that means expression 5 fills ¡missing 1¿, expression 22 fills ¡missing 2¿, and expression 3 fills ¡missing 3¿.

The question is: Find all integers $n \geq 3$ such that the following property holds: if we list the divisors of $n!$ in increasing order as $1 = d_1 < d_2 < \cdots < d_k = n!$, then we have

$$d_2 - d_1 \leq d_3 - d_2 \leq \cdots \leq d_k - d_{k-1}.$$

Find all integers $n \geq 3$ such that the following property holds: if we list the divisors of $n!$ in increasing order as $1 = d_1 < d_2 < \cdots < d_k = n!$, then we have

$$d_2 - d_1 \leq d_3 - d_2 \leq \cdots \leq d_k - d_{k-1}.$$

The solution is: We can start by verifying that ¡missing 1¿ and $n = 4$ work by listing out the factors of ¡missing 2¿ and ¡missing 3¿. We can also see that ¡missing 4¿ does not work because the terms $15, 20$, and $24$ are consecutive factors of ¡missing 5¿. Also, ¡missing 6¿ does not work because the terms ¡missing 7¿, and $9$ appear consecutively in the factors of ¡missing 8¿.

We can start by verifying that ¡missing 9¿ and ¡missing 10¿ work by listing out the factors of ¡missing 11¿ and 4!. We can also see that ¡missing 12¿ does not work because the terms $15, 20$, and $24$ are consecutive factors of 5!. Also, ¡missing 13¿ does not work because the terms ¡missing 14¿, and $9$ appear consecutively in the factors of ¡missing 15¿.

Note that if we have a prime number ¡missing 16¿ and an integer ¡missing 17¿ such that both $k$ and ¡missing 18¿ are factors of ¡missing 19¿, then the condition cannot be satisfied.

If ¡missing 20¿ is odd, then ¡missing 21¿ is a factor of ¡missing 22¿. Also, ¡missing 23¿ is a factor of $n!$. Since ¡missing 24¿ for all ¡missing 25¿, we can use Bertrand's Postulate to show that there is at least one prime number $p$ such that ¡missing 26¿. Since we have two consecutive factors of ¡missing 27¿ and a prime number between the smaller of these factors and $n$, the condition will not be satisfied for all odd ¡missing 28¿.

If ¡missing 29¿ is even, then $(2)(\frac{n-2}{2})(n - 2) = n^2 - 4n + 4$ is a factor of ¡missing 30¿. Also, $(n - 3)(n - 1) = n^2 - 4n + 3$ is a factor of ¡missing 31¿. Since ¡missing 32¿ for all $n \geq 8$, we can use Bertrand's Postulate again to

---

show that there is at least one prime number $p$ such that ¡missing 33¿. Since we have two consecutive factors of ¡missing 34¿ and a prime number between the smaller of these factors and $n$, the condition will not be satisfied for all even ¡missing 35¿.
Therefore, the only numbers that work are $n = 3$ and ¡missing 36¿.
The formulae are:

- ¡expression 1¿ $n = 6$

- ¡expression 2¿ $n = 5$

- ¡expression 3¿ $3!$

- ¡expression 4¿ $k + 1$

  ... (omitted for brevity)

- ¡expression 33¿ $n < p < n^2 - 4n + 3$

- ¡expression 34¿ $p > n$

- ¡expression 35¿ $n < p < n^2 - 2n$

- ¡expression 36¿ $n = 4$

Your final answer should be STRICTLY in the format:
**Detailed reasoning**
Answer: ¡comma separated list of numbers representing expression identifiers¿

## C. Additional Technical Details

### C.1. Inference Prompts

For questions from the MATH and AIME benchmarks, we use the following prompt.

> **MATH and AIME Prompt**
>
> Please answer the following question. Think carefully and in a step-by-step fashion. At the end of your solution, put your final result in a boxed environment, e.g. $\boxed{42}$.
> The question would be here.

For questions from the LiveBench Math and LiveBench Reasoning benchmarks, which already come with their own instructions and formatting requests, we do not provide any accompanying prompt and simply submit the model the question verbatim.

> **LiveBench Prompt**
>
> The question would be here.

### C.2. LM-Based Scoring

Given a tuple consisting of a question, ground-truth solution, and candidate response, we grade the correctness of the candidate response by querying a Gemini-v1.5-Pro-002 model to compare the candidate and ground-truth solutions. This involves repeating the following process five times: (1) send a prompt to the model that provides the question, the correct ground-truth solution, and the candidate response, and asks the model to deliberate on the correctness of the candidate response; and (2) send a followup prompt to the model to obtain a correctness ruling in a structured format. If a strict majority of (valid) responses to the second prompt evaluate to a JSON object with the key-value pair $\boxed{\text{"student\_final\_answer\_is\_correct" = True}}$ rather than $\boxed{\text{"student\_final\_answer\_is\_correct" = False}}$, the candidate response

is labeled correct. Otherwise, the candidate response is labeled incorrect. These queries are all processed with temperature zero. The prompts, which can be found at the end of this subsection, ask the language model to (1) identify the final answer of the given response, (2) identify the final answer of the reference (ground truth) response, and (3) determine whether the final answer of the given response satisfactorily matches that of the reference response, ignoring any non-substantive formatting disagreements. In line with convention, we instruct our scoring system to ignore the correctness of the logic used to reach the final answer and rather only judge the correctness of the final answer. The model is asked to label all non-sensical and incomplete responses as being incorrect.

As a form of quality assurance, every scoring output for the Consistency@200 and Verification@200 figures depicted in Table 1 was manually compared against human scoring. No discrepancies between automated and human scoring were found on the MATH and AIME datasets for both Consistency@200 and Verification@200. No discrepancies were found on LiveBench Reasoning for Consistency@200. For Verification@200, one false positive (answer labeled by automated system as being incorrect but labeled by human as being correct) and one false negative (answer labeled by automated system as being correct but labeled by human as being incorrect) were identified on LiveBench Reasoning; three false positives and four false negatives were identified on LiveBench Math. For Consistency@200, two false negatives were identified on LiveBench Math. This means that LM scoring matched human scoring 99% of the time, and the choice of human versus automated scoring matters little to our results.

---

**Prompt 1**

You are an accurate and reliable automated grading system. Below are two solutions to a math exam problem: a solution written by a student and the solution from the answer key. Your task is to check if the student's solution reaches a correct final answer.
Your response should consist of three parts. First, after reading the question carefully, identify the final answer of the answer key's solution. Second, identify the final answer of the student's solution. Third, identify whether the student's final answer is correct by comparing it to the answer key's final answer.
# The question, answer key, and student solution
The math exam question:
```
The question would be here.
```

The answer key solution:
```
The reference solution would be here.
```

The student's solution:
```
The candidate solution would be here.
```

# Your response format
Please structure your response as follows. PROVIDE A COMPLETE RESPONSE.
```
# Answer Key Final Answer
Identify the final answer of the answer key solution. That's all you need to do here: just identify the final answer.
A "final answer" can take many forms, depending on what the question is asking for; it can be a number (e.g., "37"), a string (e.g., "ABCDE"), a sequence (e.g., "2,3,4,5"), a letter (e.g., "Y"), a multiple choice option (e.g. "C"), a word (e.g., "Apple"), an algebraic expression (e.g. "$x^2 + 37$"), a quantity with units (e.g. "4 miles"), or any of a number of other options. If a solution concludes that the question is not answerable with the information provided or otherwise claims that there is no solution to the problem, let the final answer be "None". If the solution does not produce any final answer because it appears to be cut off partway or is otherwise non-sensical, let the solution's final answer be "Incomplete solution" (this could only ever possibly happen with the student solution).
YOUR RESPONSE HERE SHOULD BE BRIEF. JUST IDENTIFY WHAT THE QUESTION IS ASKING FOR, AND IDENTIFY THE ANSWER KEY'S FINAL ANSWER. DO NOT ATTEMPT TO ANSWER THE QUESTION OR EVALUATE INTERMEDIATE STEPS.

# Student Solution Final Answer
Identify the final answer of the student solution.
YOUR RESPONSE HERE SHOULD BE BRIEF. JUST IDENTIFY WHAT THE QUESTION IS ASKING FOR, AND IDENTIFY THE STUDENT'S FINAL ANSWER. DO NOT ATTEMPT TO ANSWER THE QUESTION OR EVALUATE INTERMEDIATE STEPS.
# Correctness
Simply evaluate whether the student's final answer is correct by comparing it to the answer key's final answer. Compare the student's final answer against the answer key's final answer to determine if the student's final answer is correct.
* It does not matter how the student reached their final answer, so long as their final answer itself is correct.
* It does not matter how the student formatted their final answer; for example, if the correct final answer is $\boxed{7/2}$, the student may write ***3.5*** or $\boxed{\text{three and a half}}$ or $\boxed{\frac{14}{4}}$. It does not matter if the student's final answer uses the same specific formatting that the question asks for, such as writing multiple choice options in the form "(E)" rather than "***E***".
* It does not matter if the student omitted units such as dollar signs.
* If the student solution appears to be truncated or otherwise incoherent, e.g. due to a technical glitch, then it should be treated as being incorrect.
ONCE AGAIN, DO NOT EVALUATE INTERMEDIATE STEPS OR TRY TO SOLVE THE PROBLEM YOURSELF. THE ANSWER KEY IS ALWAYS RIGHT. JUST COMPARE THE FINAL ANSWERS. IF THEY MATCH, THE STUDENT ANSWER IS CORRECT. IF THEY DO NOT MATCH, THE STUDENT ANSWER IS INCORRECT.
# Summary
* Answer key final answer: (The final answer of the answer key solution. Please remove any unnecessary formatting, e.g. provide "3" rather than "$\boxed{3}$", provide "E" rather than "***E***", provide "1, 2, 3" rather than "[1, 2, 3]", provide "4 ounces" rather than "4oz".)
* Student final answer: (The final answer of the student's solution. Please remove any unnecessary formatting, e.g. provide "3" rather than "$\boxed{3}$", provide "E" rather than "***E***", provide "1, 2, 3" rather than "[1, 2, 3]", provide "4 ounces" rather than "4oz".)
* Student final answer is correct?: (Does the student final answer match the answer key final answer? Please provide "true" or "false".)
```

## Prompt 2

Please structure your output now as JSON, saying nothing else. Use the following format: ``` { "answer_key_final_answer": str (the final answer of the answer key solution; please remove any formatting"), "student_final_answer": str (the final answer of the student's solution; please remove any formatting"), "student_final_answer_is_correct": true/false, }

### C.3. Implementation of Consistency@k

Consistency@k measures the performance of a model by evaluating the correctness of the most common answer reached by the model after being run $k$ times. An important consideration with implementing consistency@k is that there are many choices for the equivalence relation one can use to define "the most common answer". We define two candidate responses as reaching the same answer if their final answer is the same. We determine a candidate response's final answer by prompting a language model to identify the final answer from the candidate response; we then strip the extracted final answer of leading and trailing whitespace. We determine equivalence with a literal string match. After determining the most common final answer to a question, we use the string $\boxed{\text{"The final answer is \{final answer\}"}}$ as the consistency@k response. Note that we could have instead randomly chosen a candidate response corresponding to the most common final answer, and used that selected response as the consistency@k response—we have found that, because our LM-based scoring system evaluates correctness using only the final answer, this alternative results in the same consistency@k metrics.

## C.4. Benchmark Evaluation Prompts

The benchmark performances reported in Table 5 are obtained with the following prompts. The following prompt is used for the comparison task.

---

**Comparison Task Prompt Part 1**

Question here.
Here are two solutions to the above question. You must determine which one is correct. Please think extremely carefully. Do not leap to conclusions. Find out where the solutions disagree, trace them back to the source of their disagreement, and figure out which one is right.
Solution 1:
First solution here.
Solution 2:
Second solution here.

---

**Comparison Task Prompt Part 2**

Now summarize your response in a JSON format. Respond in the following format saying nothing else:
{
"correct_solution": 1 or 2
}

---

The following prompt is used for the scoring task.

---

**Scoring Task Prompt Part 1**

Question here.
I include below a student solution to the above question. Determine whether the student solution reaches the correct final answer in a correct fashion; e.g., whether the solution makes two major errors that still coincidentally cancel out. Please be careful and do not leap to conclusions without first reasoning them through.
Solution:
Solution here.

---

**Scoring Task Prompt Part 2**

Now summarize your response in a JSON format. Respond in the following format saying nothing else:
{
"is_solution_correct": 'yes' or 'no'
}

---

# D. Self-Verification Prompts

In this section we present the prompts used to query models to self-verify candidate responses. In Table 4, this corresponds to the "Main" prompt style. We also provide the prompts used for tie-breaking comparisons. These prompts are broken into multiple parts to encourage longer responses.

## D.1. Verification Scoring Prompts

---

**Verification Prompt 1**

[Commandments] Over the coming interactions, you must fulfill the following commandments: 1. *Be excruciatingly detailed and exhaustive in your analyses.* This will often mean that your responses will be long. Do not cut your responses short. When you are asked to fulfill a list of tasks, you must fulfill each and every task to completion.
2. *Be structured and systematic in your responses*. Organize your thoughts in a clear hierarchical format. Use neutral, rigorous mathematical language to write things in your own words and avoid subjective descriptions.
3. *Never put the cart before the horse.* Before making a claim or statements, always verbally reason out your chain of thought and convince yourself of it in an exhaustive fashion. Instead of saying "X is Y because Z", say "Consider Z. Therefore ..., meaning that X is Y.". Avoid premature conclusions.
[Background] An examiner has presented a math problem along with a "candidate solution". Your purpose is to assist in evaluating whether the *final answer* in the candidate solution is correct. We will guide you through a series of exercises to fulfill this purpose. The question is a real exam problem, which means there is always one unique correct answer.
Here is the math question. {} Here is the candidate solution. {}
** Current Task ** Your task is to rewrite the candidate solution into a rigorous, structured format consistent with mathematical convention. You should rewrite the solution in your own words, and work out every step in exhaustive detail, allowing us to more easily check for errors down the line. As a starting point, you will first decompose the candidate solution into a list of self-contained *lemmas*. Later on, we will have you use your lemmas to rewrite the candidate solution in the following format:

```
\begin{theorem} ... is \boxed{...}. \end{theorem}
\begin{proof}[Proof of Theorem] ... \end{proof}
\begin{lemma}\label{...} ... \end{lemma}
\begin{proof}[Proof of Lemma~\ref{...}] ... \end{proof}
...
```

For now, focus only on identifying, structuring, and clearly defining each lemma without proofs. We will write proofs for these lemmas, and a proof for the final answer using these lemmas, at a later point in time.
[Steelmanning] If you encounter any part of the candidate solution that seems incorrect or unclear, reason about it in your scratchpad. Explain your thought process to clarify your understanding. Consider whether there might be an alternative interpretation or if you could be overlooking or overcomplicating something. If, after reasoning through it, you still find it confusing, note in your scratchpad that this part may need to be revisited and proceed while treating the potentially problematic step as temporarily correct. At the current stage, we want to try to presume the candidate solution as being correct. Try your best to *steel man* the candidate solution.
[Task Specifics] To ensure accuracy, you will decompose the candidate solution into self-contained *lemmas*. These lemmas will serve as building blocks, allowing us to reconstruct the candidate solution in a more rigorous and conventional mathematical format.
[Steps to Follow] 1. **Initial Analysis**: Carefully read through the candidate solution and break it down into its primary steps. 2. **Lemma Identification**: Identify each logical segment of the candidate solution that could serve as a lemma. For each lemma: - **Scratchpad**: Before writing the lemma formally, record a brief scratchpad of thoughts outlining assumptions, conditions, and any definitions necessary to ensure the lemma is self-contained. This should help you catch potential errors or misapplications that might be subtle. Reason about how to write the lemma in a fashion that allows for easy verification of both the lemma, and how the lemma is used in the final proof. - **Lemma Statement**: After organizing your thoughts, write each lemma in a clear and rigorous manner. Ensure each lemma is isolated from others, so it does not depend on assumptions not explicitly stated within it.
[Response Format] Structure your response as follows.

```
# Analyzing the solution
(Study the language of the question, keeping an eye out for important
details.)
(Identify the candidate solution's final answer. Work backwards through the
solution to identify the intermediate claims that led to the final answer.)
```

---

```
(Proceed from the beginning of the candidate solution, working forwards to
identify the main steps of the solution.)
# Identifying lemmas
Scratchpad: [Reason about what lemma you should write. Reason about how to
word the statement of the lemma to make it easy to verify when the lemma is
being used correctly. Reason about how to write the lemma in a
self-contained fashion, and what definitions and assumptions are needed.]
\begin{lemma}
\label{lem:lemma_name}
[State the lemma formally here]
\end{lemma}
Scratchpad: ...
\begin{lemma}
...
\end{lemma}
...
```

### Verification Prompt 2

Now that you have structured the candidate solution into lemmas, you will proceed to rewrite the candidate solution in your own words. We will first have you write the *main proof for the final answer*. At the next round, I will ask you to write proofs for each lemma you identified.

[Details on Writing a **Main Proof for Final Answer**] Write a rigorous "main proof" that directly addresses the candidate solution's final answer. This proof should reference each lemma sequentially to logically reach the final answer. Ensure that the proof is rigorous and that each step explicitly relies on or follows from a lemma. Your goal is to work out every baby step of the proof in exhaustive detail, spelling out every tiny step, including even tiny details about arithmetic and algebraic manipulations.

[Objective] Your rewrite of the candidate solution should be comprehensive and precise. Your goal is to write the candidate solution so that it is easy to verify whether it is correct or identify any errors at both the high and low levels.

[Steelmanning] Once again, try to "steel man" the candidate solution. If you encounter any part of the candidate solution that seems incorrect or unclear, reason about it in your scratchpad and try to find an explanation; then proceed treating the potentially problematic step as correct. We want to try to presume the candidate solution as being correct for now.

[Response Format] Structure your response as follows to ensure clarity.

```
\begin{theorem}[Main Claim]
(Write the math question and final answer as a statement. For example, "The
smallest non-zero integers is \boxed{1}.". This statement should contain all
information from the question, and be written in your own words. If it is
not obvious how to write the student solution's final answer as a theorem,
you can simply let the theorem take the form: "{Question}? The answer is
{Answer}.")
\end{theorem}
Scratchpad: (Reason about how you will write the main proof.)
\begin{proof}
(Provide a proof of the final result. Remember to explicitly reference
lemmas via \ref{lem:lemma_name}.)
\end{proof}
```

---

### Verification Prompt 3

Your task is now to provide proofs for each and every lemma you identified in the candidate solution.

[Details on Writing **Lemma Proofs**] For each lemma, write a detailed and thorough proof of the lemma. Each proof should methodically use the assumptions, definitions, and conditions in the lemma's statement to verify the conclusion. This proof should be rigorous and work out every baby step in exhaustive detail, spelling out every tiny step, including even tiny details about arithmetic and algebraic manipulations. Remember that you may encounter a lemma that "builds" on other lemmas, in that it uses the other lemmas to prove its claim. In these cases, you should have already written the lemma so that it incorporates those other lemmas as "assumptions", so that you do not need to replicate the work. If this is not the case, please revise the lemma statement accordingly.

[Objective] Your rewrite of the candidate solution should be comprehensive and precise. Your goal is to write the candidate solution so that it is easy to verify whether it is correct or identify any errors at both the high and low levels.

[Steelmanning] Once again, try to "steel man" the candidate solution. If you encounter any part of the candidate solution that seems incorrect or unclear, reason about it in your scratchpad and try to find an explanation; then proceed treating the potentially problematic step as correct. We want to try to presume the candidate solution as being correct for now.

[Response Format] Structure your response as follows to ensure clarity and rigor. You must provide proofs for all lemmas. Do not skip any lemmas, defer any writing to the future, or leave any proofs incomplete. You must return full proofs for every lemma; I am an automated system and unable to handle incomplete responses. You will not have a chance to finish your response later. Provide a complete response.

```
\begin{lemma}
\label{...}
...
\end{lemma}
Scratchpad: (Reason about how you will write the lemma proof.)
\begin{proof}
(Provide a proof of the lemma.)
\end{proof}
... (repeat for EVERY lemma)
```

---

### Verification Prompt 4

Now we will proceed to analyzing the correctness of each suspicious claim. For each item in the claim: 1. Check if the candidate solution's final answer indeed depends on the claim. This step is important to rule out false positives where the candidate solution makes a False claim, but the claim does not really affect the final answer so it's Falsity does not result in an actual error in the final answer. If this check is failed, i.e. the final answer does not depend on the claim, then discard this claim and continue to the next suspicious claim. 2. Check that you are unable to prove the suspicious claim is True. Do this by trying to prove the suspicious claim. You can make multiple attempts. If you are able to prove the suspicious claim, meaning that the suspicious claim is correct, then discard this claim and continue to the next suspicious claim. 3. Check that the claim is False by proving a corrected alternative version of the claim. If you are unable to do this, continue to the next suspicious claim. 4. Correct this suspicious claim in the candidate solution and determine the corrected final answer in a step-by-step manner. If the final answer changes or correcting this error invalidates the candidate solution's proof approach, mark this suspicious claim as a "fatal error". At the end, say **Yes** if you found a fatal error. Otherwise, say **No, final answer is correct**.

---

### Verification Prompt 5

Now we will proceed to analyzing the correctness of the candidate solution. We have two goals: identify if there is an error, and if there is an error, repair the candidate solution and identify whether the final answer has changed. For now, we will focus on methodically combing through the candidate solution and checking each step for a potential error.

[Steps to Follow] You will proceed through the candidate solution, one baby step at a time. In your exhaustive investigation of the solution, you will first form a short list of potential errors. Each entry in this list should be written as a self-contained, standalone mathematical claim that the candidate solution relies on being correct, but which you find suspicious upon first inspection. After forming this list, you will then proceed to validate each potential error by performing a detailed error validation check.

* Main Proof * For now, focus on the main proof. Assume that every lemma, as written, is correct. Step through the main proof of the candidate solution in exhaustive detail to find potential errors. Do this by first proceeding through the main proof one sentence (or other reasonable unit of content) at a time. Quote the part of the proof you are at. Then add a discussion where you try to elaborate on that part and verify every baby step made in that unit of the proof. Then repeat this for the next unit of the proof until you have gone through (quoted and analyzed) each part of the proof. You should structure your output like

```
> Quoted part of the proof
Your analysis and detailed discussion verifying every small thing, e.g.
redoing every baby step, checking assumptions, rephrasing things to make
sure they sound correct to you, etc. Make sure you double check the exact
language of the question and the exact language of the lemmas used.
> Next quoted part of the proof
...
```

Then, write down in a bullet point list any potential errors you find in the main proof. Before each bullet point, write a scratchpad that verbalizes: your thought process and—upon finding an error—your thought process for writing the erroneous step as a self-contained mathematical claim. Try to be selective and only include things that you consider likely to be errors. You should structure your output like

```
Scratchpad: ...
* ...
Scratchpad: ...
* ...
...
```

---

**Verification Prompt 6**

You are now to verify the proof of each and every lemma. You must verify all lemmas in this conversation turn; you will not have another chance.

* Lemma ... Proof * (repeat this for each lemma) Follow the same procedure as before. First, step through the lemma proof one sentence (or other reasonable unit of content) at a time. Quote the part of the proof you are at. Then add a discussion where you try to elaborate on that part and verify every baby step made in that unit of the proof. Then repeat this for the next unit of the proof until you have gone through (quoted and analyzed) each part of the proof. Then, write down in a bullet point list any potential errors you find in the lemma proof. Before each bullet point, write a scratchpad that verbalizes: your thought process and—upon finding an error—your thought process for writing the erroneous step as a self-contained mathematical claim. Try to be selective and only include things that you consider likely to be errors. Structure your response as follows.

```
# Review of Lemma ...
## Stepping through
> Quoted part of the proof
Your analysis and detailed discussion verifying every small thing, e.g.
redoing every baby step, checking assumptions, rephrasing things to make
sure they sound correct to you, etc. Make sure you double check the exact
language of the lemma's claim.
> Next quoted part of the proof
## Potential errors
```

```
Scratchpad: ...
* ...
Scratchpad: ...
* ...
...
```

## Verification Prompt 7

Now we will proceed to analyzing the correctness of each suspicious claim. For each item in the claim: 1. Check if the candidate solution's final answer indeed depends on the claim. This step is important to rule out false positives where the candidate solution makes a False claim, but the claim does not really affect the final answer so it's Falsity does not result in an actual error in the final answer. If this check is failed, i.e. the final answer does not depend on the claim, then discard this claim and continue to the next suspicious claim. 2. Check that you are unable to prove the suspicious claim is True. Do this by trying to prove the suspicious claim. You can make multiple attempts. If you are able to prove the suspicious claim, meaning that the suspicious claim is correct, then discard this claim and continue to the next suspicious claim. 3. Check that the claim is False by proving a corrected alternative version of the claim. If you are unable to do this, continue to the next suspicious claim. 4. Correct this suspicious claim in the candidate solution and determine the corrected final answer in a step-by-step manner. If the final answer changes or correcting this error invalidates the candidate solution's proof approach, mark this suspicious claim as a "fatal error". At the end, say **Yes** if you found a fatal error. Otherwise, say **No, final answer is correct**.

## Verification Prompt 8

I want you to now, in the style of the *rewritten solution* (i.e., in the theorem-lemma format I had you rewrite the original solution in), write an improved solution that does the following: 1) It corrects any errors you found in the original solution (only if you ended up finding an error). If you do not know how to fix the error, then just write the step you are unsure about as an "Assumption (to be revisited later)", making it clear that the proof is not complete and needs that step to be filled in. 2) It clarifies ambiguities or fills in gaps in the original solution that led you to suspect errors (even if the suspicion was not ultimately substantiated). We want your improved solution to be clear enough that future readers would not have to work through the same worries you did. 3) It adopts the thoroughness, structured format, exhaustive detail, and rigor of your rewritten version of the solution.

I have included again the original question and original solution below for your reference. {}{} For now, I want you to brainstorm how to write the revised solution. Do not yet proceed to writing the solution. Structure your response as follows. Do not cut your response short. You have unlimited space.

```
# What errors need to be corrected?
Discuss in detail, in a step-by-step manner, what errors you found in the
original solution and how you plan to correct them.
If you found no errors previously, note so and continue to the next section.
# Which ambiguities need to be clarified or gaps need to be filled?
Discuss in detail, in a step-by-step manner, what ambiguities or gaps you
plan to clarify.
Draw on our previous exchanges explicitly. Recall what were potential points
of confusion during this verification process.
# Action Plan
Prepare a rough battle plan for how you will write the improved solution.
Note that it will probably be longer than you original solution, given all
of the revisions you have planned. So this is just to plan out the main
parts, and things to keep an eye out for.
```

---

**Verification Prompt 9**

Now, I want you to write your improvement of the original solution. You must provide the entirety of your solution and nothing else. This message will be copy pasted to an external system, so your response must be complete and self-contained. Do not shorten anything. Provide it to me exactly and in its entirety. Remember to say nothing else and structure your output as follows, wrapped in tripe quotes and saying nothing else.

---

## D.2. Comparison Prompts

---

**Comparison Prompt 1**

Your job is to answer a difficult math question. I will provide you with two solutions written by my students. These two solutions disagree over what the answer should be. Both solutions may potentially have minor errors, even if one indeed reaches the final answer despite the minor flaws. You will be given a series of instructions, over multiple interactions, that will guide you through discerning the correct final answer. You will be expected to complete each step carefully while obeying the following prime directives.
1. Always provide complete responses. Never shorten your responses. You are allocated 10,000 tokens per-response. Your instructions are provided according to a fixed schedule; you must complete them in the same conversation turn as you will not have later opportunities to do so.
2. Speak carefully. Always reason in a step-by-step chain-of-thought manner. Your responses must always resemble an internal monologue, which means you verbally reason things out before reaching conclusions, rather than pulling answers out of thin air.
3. Be rigorous. Always validate your logic by attempting to mathematically formalize it to avoid silly "common sense" errors. Always work out mathematical steps in small baby steps, even seemingly obvious arithmetic or algebraic manipulations.
4. Backtrack when you have made a mistake. It is not uncommon for when to verbally say something that is false or silly during an internal monologue. Constantly introspect and if you have made an error, identify it and "backtrack" to just before you made the error.
5. Never claim that anything is incorrect or wrong or right or correct. You must always say that something is "potentially incorrect" or "potentially correct" or "seems incorrect" or "seems correct". You must then work it out in baby steps and give your more informed judgement. But you will never say that something is correct or right or wrong.
# The Question {}
# The First Solution {}
# The Second Solution {}
# Your current Task Examine the first solution. Focus for now on just reading through the main proof and each lemma's statement. You must read through the main proofs and lemma statements in a meticulous sentence-by-sentence manner. For now, you do not need to read through the lemma proofs in detail. Though later on, if we have questions arise about particular lemmas, you will need to review the lemma proof carefully. In addition, I want you to narrate your process of reading through each proof. This means that, while you read through a solution, you must always quote the part of the solution that you are currently reading. Then, under your quote, provide your mental process. Here are some questions that you should keep top of mind: What is the approach being taken by the solution? What are the main leaps in logic? How does the solution try to be rigorous? Structure your response as follows:

```
# Main Proof
> "Quote of the sentence you are reading."
Your thought process. Discuss the quote. Perhaps compare it to a sentence in
another solution that is doing the same exact thing. Perhaps note that it
does a fairly complex algebraic manipulation, and spend a minute to
double-check it by working it out in baby-steps. If it references a lemma,
discuss the lemma's statement, how it's being used, and why it's allowed to
be used in this way.
```

```
> "Quote of next sentence..."
...
# Lemma 1
...
(continue through the ENTIRE REST OF THE SOLUTION)
```

Remember you only need to do this for the first solution for now. You must provide a complete response. You must go through the entire solution line by line. Ignore character limits and do not cut your response short. Do not cut your response short, you will not get another chance.

---

**Comparison Prompt 2**

Do the same for the second solution. Again, be meticulous and adopt the same output format.

---

**Comparison Prompt 3**

Identify similarities and disagreements between the first solution and the second solution. Do not yet judge which side is right; merely try to be investigative about: what are the sources of these disagreements? You must continue to "quote" any solution parts that you reference. Remember your mandates:
1. Always provide complete responses. Never shorten your responses. You are allocated 10,000 tokens per-response. Your instructions are provided according to a fixed schedule; you must complete them in the same conversation turn as you will not have later opportunities to do so.
2. Speak carefully. Always reason in a step-by-step chain-of-thought manner. Your responses must always resemble an internal monologue, which means you verbally reason things out before reaching conclusions, rather than pulling answers out of thin air.
3. Be rigorous. Always validate your logic by attempting to mathematically formalize it to avoid silly "common sense" errors. Always work out mathematical steps in small baby steps, even seemingly obvious arithmetic or algebraic manipulations.
4. Backtrack when you have made a mistake. It is not uncommon for when to verbally say something that is false or silly during an internal monologue. Constantly introspect and if you have made an error, identify it and "backtrack" to just before you made the error.
5. Never claim that anything is incorrect or wrong or right or correct. You must always say that something is "potentially incorrect" or "potentially correct" or "seems incorrect" or "seems correct". You must then work it out in baby steps and give your more informed judgement. But you will Never say that something is correct or right or wrong..
Be extremely careful with respect to the exact wording of the question: {}
Remember: you must be humble and careful in your judgements. Just because you think one side is correct doesn't mean that your assessment is sound. Always keep an open mind. Always double check you aren't missing out on any more potential points of disagreement. Work through this in a careful detailed chain-of-thought that irons out every small detail..
Structure your responses into the following sections. Each section must be a lengthy, detailed, well-structured investigation.

```
# Identify Disagreements
Reading through each solution, identify as many as places as possible where
the solutions differ. Be meticulous and form a detailed list.
# Attribute Disagreements
```

```
For each disagreement, try to trace the disagreement back to earlier places
in the respective proofs. What are the points at which the solutions begin
to diverge? Can we guess at the root causes of the disagreements? Could it
be because the solutions disagree over how to interpret the question? Could
it be because they disagree about what approach to take? Could it be because
the solutions reach a different calculation? CAREFULLY pour over solutions
from both sides. This detective work must be careful and methodical and
detailed and exhaustive.
```

You must provide your full response. Ignore character limits. Do not cut your response short. Obey my instructions exactly.

---

### Comparison Prompt 4

Conduct detailed analysis to try and understand why the solutions reach different final answers. Do not yet judge who is right. Do they disagree on how to read the question? Do they just happen to take very different mathematical approaches? Do they diverge at a particular logical step or calculation? You must continue to "quote" any solution parts that you reference. Remember your mandates:

1. Always provide complete responses. Never shorten your responses. You are allocated 10,000 tokens per-response. Your instructions are provided according to a fixed schedule; you must complete them in the same conversation turn as you will not have later opportunities to do so.

2. Speak carefully. Always reason in a step-by-step chain-of-thought manner. Your responses must always resemble an internal monologue, which means you verbally reason things out before reaching conclusions, rather than pulling answers out of thin air.

3. Be rigorous. Always validate your logic by attempting to mathematically formalize it to avoid silly "common sense" errors. Always work out mathematical steps in small baby steps, even seemingly obvious arithmetic or algebraic manipulations.

4. Backtrack when you have made a mistake. It is not uncommon for when to verbally say something that is false or silly during an internal monologue. Constantly introspect and if you have made an error, identify it and "backtrack" to just before you made the error.

5. Never claim that anything is incorrect or wrong or right or correct. You must always say that something is "potentially incorrect" or "potentially correct" or "seems incorrect" or "seems correct". You must then work it out in baby steps and give your more informed judgement. But you will never say that something is correct or right or wrong.

Be extremely careful with respect to the exact wording of the question: {}

Remember: you must be humble and careful in your judgements. Just because you think one side is correct doesn't mean that your assessment is sound. Always keep an open mind. Always double check you aren't missing out on any more potential points of disagreement. Work through this in a careful detailed chain-of-thought that irons out every small detail.. You are NOT to judge which side is correct, for now.

Structure your responses into the following sections. Each section must be a lengthy, detailed, well-structured investigation.

```
# Crystallize Approaches
If the two sets of solutions differ in approach, it can be hard to judge
which side is correct about the final answer since you cannot easily compare
the solutions of each. However, even if you cannot directly compare the
steps, you can try to find "intermediate" points of disagreement. That is,
as part of their respective arguments, two solutions that use completely
different approaches may make contradicting subclaims that you can
investigate in more detail. These disagreements may point you to where one
approach has a "mistake" in its proof. Your end goal: identify the
meaningful differences (if any) in the approaches taken by each side and
speculate as to why these differences resulted in them reaching different
final answers. Remember: you must be humble and careful in your judgements.
Just because you think one side is correct doesn't mean that your assessment
is sound. Always keep an open mind. Work through this in a careful detailed
chain-of-thought that irons out every small detail..
# Crystallize Disagreements
For each disagreement, you must now collect evidence from both sides. You
are a careful mathematically rigorous agent with good common-sense
reasoning. Try to use your common sense and rigour to work through each
side's evidence. Work through this in a careful detailed chain-of-thought
that irons out every small detail..
```

You must provide your full response. Ignore character limits. Do not cut your response short. Obey my instructions
exactly.

---

**Comparison Prompt 5**

You must now determine which of the two solutions reaches the correct final answer. Do not leap to judgement.
First, carefully consider the sources of the solutions disagreement over the final answer. Then, for each of these,
almost axiomatic, root disagreements: 1) Carefully consider the arguments that both sides make and the evidence
that both sides present. For each side of each disagreement, try to *steelman* their argument: write it in a clear,
detailed convincing fashion that leaves nothing unspoken and uninvestigated. 2) Use your own common sense and
mathematical skills to try and rule on which side is right. Do not leap to judgement. Be careful, rigorous, meticulous
and don't forget common sense. Finally, carefully reason through what the correct solution should be and which
side reached the correct final answer. Be meticulous. Structure your response as follows. This must be more detailed
and rigorous than all your previous responses, since this is a careful process.

```
# Recall Key Disagreements
...
# Disagreement 1
## Steelmanning Each Side
...
## Ruling On A Side
...
# Which Side Is Right?
...
```

You must provide your full response. Ignore character limits. Do not cut your response short. Obey my instructions
exactly.

# E. Examples of Benchmark Entries

## E.1. Example Entry from LiveBench Reasoning

> **Question**
>
> In this question, assume each person either always tells the truth or always lies. The person at the campground thinks their friend is lying. Mateo is at the aquarium. The person at the restaurant says the person at the hotel lies. Farid is at the movie theater. The person at the movie theater says the person at the campground lies. Ryan is at the shopping mall. The person at the cafe says the person at the campground lies. The person at the observatory says the person at the museum lies. The person at the museum says the person at the restaurant tells the truth. The person at the campground says the person at the airport tells the truth. The person at the campground tells the truth. Luna is at the restaurant. The person at the movie theater saw a firetruck. Isabella is at the observatory. The person at the amusement park says the person at the museum tells the truth. The person at the restaurant says the person at the shopping mall tells the truth. Liam is at the beach. Ayaan is at the airport. The person at the restaurant says the person at the beach tells the truth. Maya is at the amusement park. The person at the restaurant says the person at the aquarium lies. The person at the restaurant says the person at the movie theater lies. The person at the library says the person at the campground tells the truth. Elowen is at the hotel. Priya is at the library. The person at the gym says the person at the restaurant tells the truth. Kehinde is at the cafe. Anika is at the museum. Quan is at the campground. Soren is at the gym. Does the person at the museum tell the truth? Does the person at the restaurant tell the truth? Does the person at the movie theater tell the truth? Think step by step, and then put your answer in **bold** as a list of three words, yes or no (for example, **yes, no, yes**). If you don't know, guess.

> **Correct response**
>
> Here's how we can break down this logic puzzle:
> 1. **Quan is at the campground and tells the truth.** This is a given fact.
> 2. **Quan (at the campground) says the person at the airport tells the truth.** Since Quan tells the truth, the person at the airport (Ayaan) tells the truth.
> 3. **The person at the movie theater (Farid) says the person at the campground (Quan) lies.** Since Quan tells the truth, Farid is lying.
> 4. **The person at the restaurant (Luna) says the person at the hotel (Elowen) lies.** We don't know yet if Luna is lying or telling the truth.
> 5. **The person at the museum (Anika) says the person at the restaurant (Luna) tells the truth.** We still don't know if either of them are truthful.
> 6. **The person at the restaurant (Luna) says the person at the shopping mall (Ryan) tells the truth.** We still don't know Luna's truthfulness.
> 7. **The person at the restaurant (Luna) says the person at the beach (Liam) tells the truth.** Luna's truthfulness remains unknown.
> 8. **The person at the restaurant (Luna) says the person at the aquarium (Mateo) lies.** We still don't know about Luna.
> 9. **The person at the restaurant (Luna) says the person at the movie theater (Farid) lies.** We know Farid lies, so Luna is telling the truth in *this* statement.

> **Incorrect Response**
>
> Here's how we can solve this logic puzzle:
> 1. **Quan is at the campground and tells the truth.** This is given.
> 2. **The person at the airport (Ayaan) tells the truth.** Quan (campground) says the person at the airport tells the truth, and Quan tells the truth.
> 3. **Anika (museum) lies.** The person at the observatory (Isabella) says the person at the museum lies. Since we don't know yet whether Isabella tells the truth, this doesn't tell us anything about Anika. We'll have to come back to this.
> 4. **Luna (restaurant) lies.** The person at the restaurant says the person at the hotel (Elowen) lies. Later, the person at the restaurant says the person at the campground (Quan) lies. Since Quan tells the truth, Luna (restaurant) must be lying.
> 5. **Anika (museum) lies.** Because Luna lies, and Luna says that the person at the hotel lies, Elowen tells the truth. Because Anika says the person at the restaurant (Luna) tells the truth, and Luna actually lies, Anika (Museum) must lie.
> 6. **Farid (movie theater) lies.** The person at the movie theater says the person at the campground lies. Since Quan (campground) tells the truth, Farid (movie theater) must be lying.
> Therefore:

10. **Since Luna tells the truth about Farid, Luna *always* tells the truth.** This unlocks several other pieces of information.
11. **Luna (at the restaurant) says the person at the hotel (Elowen) lies.** Since Luna tells the truth, Elowen lies.
12. **Anika (at the museum) says the person at the restaurant (Luna) tells the truth.** Since Luna tells the truth, Anika is also telling the truth.
13. **The person at the cafe (Kehinde) says the person at the campground lies.** Since Quan tells the truth, Kehinde is lying.
Therefore:
* **Anika (museum):** Tells the truth. * **Luna (restaurant):** Tells the truth. * **Farid (movie theater):** Lies.
**yes, yes, no**

* **Anika (museum): lies** * **Luna (restaurant): lies** * **Farid (movie theater): lies**
**Answer: no, no, no**

### E.2. Example Entry from LiveBench Math

#### Question

How many positive perfect squares less than 2023 are divisible by 5? **(A)** 11 **(B)** 10 **(C)** 9 **(D)** 12 **(E)** 8 If you cannot determine the correct multiple-choice answer, take your best guess. Once you have your answer, please duplicate that letter five times in a single string. For example, if the answer is F, then write FFFFF.

#### Correct response

Let $n$ be a positive perfect square divisible by 5. Then $n = k^2$ for some positive integer $k$, and $5|n$, which implies that $5|k^2$. Since 5 is a prime number, $5|k^2$ means $5|k$. Thus, $k$ must be a multiple of 5, i.e., $k = 5m$ for some positive integer $m$. Then $n = k^2 = (5m)^2 = 25m^2$. Thus $n$ must be a multiple of 25. We are looking for the number of positive perfect squares less than 2023 that are divisible by 5. The perfect squares that are divisible by 5 must be multiples of 25. We seek the number of multiples of 25 that are perfect squares less than 2023. Since $n = 25m^2 < 2023$, we have $m^2 < \frac{2023}{25} \approx 80.92$. Thus, we are looking for perfect squares $m^2$ less than 80.92. The perfect squares are $1, 4, 9, 16, 25, 36, 49, 64, 81, \dots$. The perfect squares less than 80.92 are $m^2 = 1, 4, 9, 16, 25, 36, 49, 64$. There are 8 such values. The corresponding values of $n = 25m^2$ are $25(1) = 25$ $25(4) = 100$ $25(9) = 225$ $25(16) = 400$ $25(25) = 625$ $25(36) = 900$ $25(49) = 1225$ $25(64) = 1600$ The next multiple of 25 that is a perfect square would be $25(81) = 2025$, which is

#### Incorrect Response

Let $n$ be a positive integer such that $n^2$ is divisible by 5. Then $n^2$ is divisible by 5. Since 5 is a prime number, $n$ must also be divisible by 5. Thus $n$ can be written in the form $n = 5k$ for some integer $k$. We want to find how many positive perfect squares $n^2$ are less than 2023. Since $n = 5k$, we have $n^2 = (5k)^2 = 25k^2 < 2023$. Dividing by 25 we get

$$k^2 < \frac{2023}{25} \approx 80.92$$

We are looking for integer values of $k$ such that $k^2 < 80.92$. Since $k$ must be positive, $k$ can take values $1, 2, 3, \dots$. We have

- $1^2 = 1 < 80.92$

- $2^2 = 4 < 80.92$

- $3^2 = 9 < 80.92$

- $4^2 = 16 < 80.92$

- $5^2 = 25 < 80.92$

not less than 2023. So, there are 8 perfect squares less than 2023 that are divisible by 5.
Final Answer: The final answer is $\boxed{EEEEE}$

- $6^2 = 36 < 80.92$

- $7^2 = 49 < 80.92$

- $8^2 = 64 < 80.92$

- $9^2 = 81 \not< 80.92$

Thus $k$ can take values from 1 to 8. The corresponding $n$ values are $n = 5k = 5, 10, 15, 20, 25, 30, 35, 40$. The perfect squares less than 2023 and divisible by 5 are $5^2, 10^2, 15^2, 20^2, 25^2, 30^2, 35^2, 40^2$ or $25, 100, 225, 400, 625, 900, 1225, 1600$. There are 8 such numbers.
Final Answer: The final answer is $\boxed{C}$ CCCC

### E.3. Example Entry from MATH

**Question**

If 4 daps = 7 yaps, and 5 yaps = 3 baps, how many daps equal 42 baps?

**Correct response**

Let 'd' represent daps, 'y' represent yaps, and 'b' represent baps. We are given: 4 daps = 7 yaps, which can be written as: 4d = 7y
and
5 yaps = 3 baps, which can be written as: 5y = 3b
We want to find the number of daps that are equal to 42 baps. Let 'x' be the number of daps. We want to find x such that: xd = 42b
From 5y = 3b, we can express yaps in terms of baps: y = (3/5)b
Now substitute this value of y into the first equation, 4d = 7y: 4d = 7 * (3/5)b 4d = (21/5)b
We want to find xd = 42b, so we can set: xd = 42b
We have 4d = (21/5)b, so we can solve for d: d = (21/20)b
Now substitute this into xd = 42b: x * (21/20)b = 42b
Since b is not zero, we can divide both sides by b: x * (21/20) = 42
Now, solve for x: x = 42 * (20/21) x = (42/21) * 20 x = 2 * 20 x = 40
So, 40 daps are equal to 42 baps.
$\boxed{40}$ Final Answer: The final answer is $\boxed{40}$

**Incorrect Response**

Let 'd' represent daps, 'y' represent yaps, and 'b' represent baps. We are given two relationships:
1. 4 daps = 7 yaps, which can be written as $4d = 7y$
2. 5 yaps = 3 baps, which can be written as $5y = 3b$
We want to find how many daps are equal to 42 baps. We want to find $d$ in terms of $b$, and then evaluate for $b = 42$.
First, we can find the relationship between yaps and baps from the second equation: $5y = 3b$, so $y = \frac{3}{5}b$
Now, substitute this expression for $y$ in terms of $b$ into the first equation: $4d = 7y$ $4d = 7\left(\frac{3}{5}b\right)$ $4d = \frac{21}{5}b$
Now, we want to find $d$ in terms of $b$, so divide both sides by 4: $d = \frac{21}{5 \cdot 4}b$ $d = \frac{21}{20}b$
Finally, we want to find how many daps equal 42 baps, so we substitute $b = 42$: $d = \frac{21}{20} \times 42$ $d = \frac{21 \times 42}{20}$ $d = \frac{882}{20}$ $d = \frac{441}{10}$ $d = 44.1$
So, 44.1 daps equal 42 baps.
Final Answer: The final answer is $\boxed{44.1}$

### E.4. Example of Response with Invalid Reasoning but Correct Final Answer

---

**Question**

You have seven bags of gold coins. Each bag has the same number of gold coins. One day, you find a bag of 53 coins. You decide to redistribute the number of coins you have so that all eight bags you hold have the same number of coins. You successfully manage to redistribute all the coins, and you also note that you have more than 200 coins. What is the smallest number of coins you could have had before finding the bag of 53 coins?

---

**Incorrect Response Reaching Correct Final Answer**

Let $n$ be the number of coins in each of the seven bags. After finding the bag of 53 coins, you have a total of $7n + 53$ coins. You redistribute these coins into 8 bags such that each bag contains the same number of coins. Therefore, the total number of coins is divisible by 8, so $7n + 53$ is divisible by 8. $7n + 53 \equiv 0 \pmod 8$. We want to find $n$. $7n \equiv -53 \pmod 8$. Since $-53 \equiv -53 + 8(7) \equiv -53 + 56 \equiv 3 \pmod 8$, we have $7n \equiv 3 \pmod 8$.
Since $7 \equiv -1 \pmod 8$, we multiply both sides by $-1$ to get $-n \equiv -7n \equiv -3 \pmod 8$, so $n \equiv -3 \equiv 5 \pmod 8$. Thus $n$ must be of the form $8k + 5$ for some integer $k \geq 0$. The total number of coins originally was $7n$. Since $n = 8k + 5$, the total number of coins originally is $7(8k + 5) = 56k + 35$. We are given that the total number of coins is greater than 200. So $7n > 200$, or $n > \frac{200}{7} \approx 28.57$. Since $n = 8k + 5$, we want the smallest integer $k$ such that $8k + 5 > \frac{200}{7}$. For $k = 0$, $n = 5$, so $7n = 35 < 200$. For $k = 1$, $n = 13$, so $7n = 91 < 200$. For $k = 2$, $n = 21$, so $7n = 147 < 200$. For $k = 3$, $n = 29$, so $7n = 203 > 200$.
We check that $7n + 53 = 203 + 53 = 256$ which is divisible by 8 ($256 = 8 \times 32$). Then $256/8 = 32$ coins in each bag.
Thus the smallest number of coins you could have had before finding the bag is $7n = 7(29) = 203$.
Final Answer: The final answer is $\boxed{203}$.

---

## F. Examples of Solution Rewrites

### F.1. AIME Example

---

**Question**

Find the number of rectangles that can be formed inside a fixed regular dodecagon ( 12 -gon) where each side of the rectangle lies on either a side or a diagonal of the dodecagon. The diagram below shows three of those rectangles.
[asy] unitsize(0.6 inch); for(int i=0; i¡360; i+=30)  dot(dir(i), 4+black); draw(dir(i)–dir(i+30));  draw(dir(120)–dir(330)); filldraw(dir(210)–dir(240)–dir(30)–dir(60)–cycle, mediumgray, linewidth(1.5)); draw((0,0.366)–(0.366,0), linewidth(1.5)); [/asy]

---

**Correct Answer**

The solution is $\boxed{315}$.

---

**Candidate Response**

Let the vertices of the regular dodecagon be labeled as $V_1, V_2, \ldots, V_{12}$. Each side of the rectangle lies on either a side or a diagonal of the dodecagon.
Consider two parallel diagonals (or sides). They determine the sides of the rectangle parallel to them. We need to choose two parallel diagonals (or sides) to form a rectangle.
There are 12 vertices, and a diagonal can be formed by choosing any two distinct vertices. The number of diagonals and sides is $\binom{12}{2} = \frac{12 \times 11}{2} = 66$. However, there are 12 sides. So there are $66 - 12 = 54$ diagonals. Two parallel diagonals or sides can be drawn as follows: 1. Diagonals parallel to sides: 12 sides so $12/2 = 6$ pairs 2. Diagonals

---

spanning 2 vertices: $\binom{12}{4}/2$ sets of 4 vertices, so we divide by 2 to get distinct pairs. $\binom{12}{4}/2 = \frac{12 \cdot 11 \cdot 10 \cdot 9}{4 \cdot 3 \cdot 2 \cdot 2} = 495/2$, which is not an integer. Something is wrong. The pairs of diagonals can only be separated by $1, 2, 3, 4, 5$ vertices on either side. Let $n$ be the number of vertices. For $n = 12$, 1 vertex apart: 12 diagonals. There are $12/2 = 6$ pairs. 2 vertices apart: 12 diagonals. $12/2 = 6$ pairs 3 vertices apart: 12 diagonals. $12/2 = 6$ pairs. 4 vertices apart: 12 diagonals. $12/2 = 6$ pairs. 5 vertices apart: 12 diagonals. $12/2 = 6$ pairs. So there are $5 \times 6 + 6$ pairs of parallel diagonals, which is 36 pairs. Number of rectangles is the number of pairs of parallel segments, including sides. For a regular $n$-gon, the number of pairs of parallel diagonals or sides is $n\lfloor (n-1)/2 \rfloor$. For $n = 12$, it's $12(5) = 60$ segments. For each length, there are 12 diagonals. Length can range from 1 side to 5 sides. $1 + 2 + 3 + 4 + 5 + 6 = \binom{12}{4} = 495$.

Consider 12 sides as 12 possible segments. Parallel to each are 5 lengths of segments. So total parallel segments $12 \cdot 5 + 12/2 = 60 + 6 = 66$, incorrect. Each vertex pairs with 5 other vertices giving the same direction. So we have $12 \times 5/2 = 30$ pairs of directions. Given one pair of parallel directions, we can choose 6 segments in that direction. So $6^2 = 36$ possibilities. $30 \times 36 = 1080$. We are forming $\binom{12}{4} = 495$ parallelograms. However, we are considering parallel segments. The number of rectangles is $12 \times 5 \times (12/2)$.

Number of rectangles is $\frac{12^2(12-1)}{4}\frac{1}{2}\frac{1}{2} = 594$ for each parallelogram $594/3 = 198$. $\boxed{198}$.

Final Answer: The final answer is $\boxed{198}$

---

**Rewritten Candidate Response**

**Theorem F.1** (Main Claim). *The number of rectangles that can be formed inside a fixed regular dodecagon, where each side of the rectangle lies on either a side or a diagonal of the dodecagon, is* $\boxed{198}$.

*Proof.* By Lemma F.2, a regular dodecagon has 12 sides. By Lemma F.3, a regular dodecagon has $\binom{12}{2} - 12 = \frac{12 \times 11}{2} - 12 = 66 - 12 = 54$ diagonals. By Lemma F.4, two line segments formed by connecting vertices of a dodecagon are considered parallel under certain conditions.

The candidate solution calculates a value of 594. By Lemma F.6, this value is obtained by calculating $\frac{12^2(12-1)}{4}\frac{1}{2}\frac{1}{2}$. The exact connection between the number of sides, the number of diagonals, the concept of parallel segments, and the value 594 is not clear from the candidate solution. However, according to Lemma F.7, the final answer of 198 is obtained by dividing 594 by 3. Therefore, we perform this division: $594/3 = 198$.

Thus, the candidate solution concludes that the number of rectangles is 198. $\square$

**Lemma F.2.** *A regular $n$-gon has $n$ sides.*

*Proof.* An $n$-gon is defined as a polygon with $n$ sides. A regular $n$-gon is also a polygon with $n$ sides. Therefore, a regular $n$-gon has $n$ sides. $\square$

**Lemma F.3.** *A regular $n$-gon has $\binom{n}{2} - n$ diagonals.*

*Proof.* A regular $n$-gon has $n$ vertices. Any two distinct vertices can be connected by a line segment. The number of ways to choose 2 vertices out of $n$ is given by the combination formula $\binom{n}{2} = \frac{n(n-1)}{2}$.

The sides of the $n$-gon are formed by connecting adjacent vertices. There are $n$ such pairs of adjacent vertices, corresponding to the $n$ sides of the $n$-gon.

A diagonal is a line segment connecting two non-adjacent vertices. Therefore, the number of diagonals is the total number of pairs of vertices minus the number of sides: $\binom{n}{2} - n = \frac{n(n-1)}{2} - n$. $\square$

**Lemma F.4.** *Let $D$ be a regular dodecagon. Two line segments formed by connecting vertices of $D$ are considered parallel if they are either coincident sides of $D$, or if they are distinct and are both parallel to some side of $D$.*

*Proof.* This lemma is a definition, so no proof is required. $\square$

**Lemma F.5.** *Let $D$ be a regular dodecagon. The number of unordered pairs of parallel segments formed by connecting vertices of $D$ (including sides) is denoted by $P$.*

*Proof.* This lemma is a definition, introducing the variable $P$ to represent the number of unordered pairs of parallel segments. No proof is required. $\square$

**Lemma F.6.** *The value 594 is obtained via the calculation* $\frac{12^2(12-1)}{4}\frac{1}{2}\frac{1}{2}$.

*Proof.* We evaluate the expression: $\frac{12^2(12-1)}{4}\frac{1}{2}\frac{1}{2} = \frac{144 \times 11}{4} \times \frac{1}{4} = \frac{1584}{16} = \frac{792}{8} = \frac{396}{4} = \frac{198}{2} = 99 \times \frac{1}{2} \times \frac{1}{2}$
Evaluating $\frac{144 \times 11}{16}$ yields $\frac{1584}{16} = 99$. However, $99 \times \frac{1}{4} = \frac{99}{4} \neq 594$. The expression provided appears to have a calculation error, giving 99 instead of 594.
The candidate solution has made a calculation error. $12^2(12-1) = 144 * 11 = 1584$. $1584/4 = 396$. $396/2 = 198$. $198/2 = 99$. The calculation gives 99, not 594. $\qquad\square$

**Lemma F.7.** *The candidate solution's final answer is obtained by dividing 594 by 3.*

*Proof.* Dividing 594 by 3, we get $594/3 = 198$. $\qquad\square$

## F.2. LiveBench Reasoning Example

### Question

In this question, assume each person either always tells the truth or always lies. Elowen is at the hotel. Max is at the park. Grace is at the ice skating rink. The person at the barbershop says the person at the library tells the truth. Charlie is at the library. Tala is at the campground. Farid is at the theater. The person at the hotel saw a firetruck. The person at the vineyard says the person at the campground lies. The person at the farm says the person at the movie theater lies. The person at the movie theater says the person at the park lies. The person at the park says the person at the hotel tells the truth. The person at the hotel says the person at the library tells the truth. Jaxon is at the skate park. Theo is at the barbershop. The person at the airport tells the truth. The person at the bowling alley thinks their neighbor is telling the truth. The person at the ice skating rink says the person at the bowling alley tells the truth. Anika is at the airport. The person at the bookstore says the person at the vineyard lies. The person at the campground says the person at the movie theater tells the truth. The person at the restaurant says the person at the campground tells the truth. The person at the theater says the person at the park lies. Hiroshi is at the school. Zara is at the bowling alley. The person at the bowling alley says the person at the school tells the truth. The person at the park thinks their neighbor is telling the truth. The person at the skate park lies. The person at the bowling alley saw a firetruck. The person at the observatory lies. Liam is at the restaurant. The person at the library says the person at the observatory tells the truth. The person at the bowling alley says the person at the vineyard lies. Beatriz is at the observatory. Priya is at the movie theater. Nia is at the farm. Ryan is at the vineyard. Isabella is at the bookstore. Does the person at the ice skating rink tell the truth? Does the person at the bowling alley tell the truth? Does the person at the vineyard tell the truth? Think step by step, and then put your answer in **bold** as a list of three words, yes or no (for example, **yes, no, yes**). If you don't know, guess.

### Correct Answer

$\boxed{yes, yes, no}$

### Candidate Response

Here's how we can break down this logic puzzle:
1. **Start with a known truth-teller or liar:** Jaxon is at the skate park and lies.
2. **Use Jaxon's lie to find more information:** Since Jaxon lies, the person at the bowling alley, Zara, *cannot* be telling the truth. Zara says the person at the school tells the truth, but Zara is lying so Hiroshi at the school lies. Zara also says the person at the vineyard lies. Since Zara lies, Ryan at the vineyard must tell the truth.
3. **Focus on the vineyard:** Ryan, a truth-teller at the vineyard, says Tala at the campground lies.
4. **Follow Tala's lie:** Tala lies and says the person at the movie theater tells the truth. This means Priya at the movie theater *lies*.
5. **Use Priya's lie:** Priya at the movie theater lies and says Max at the park lies. Therefore, Max tells the truth.
6. **Use Max's truth:** Max at the park tells the truth and says Elowen at the hotel tells the truth.

7. **Use Elowen's truth:** Elowen tells the truth and says the person at the library tells the truth. This means Charlie at the library tells the truth.

8. **Use Charlie's truth:** Charlie tells the truth and says Beatriz at the observatory lies. This is confirmed by Jaxon's lie - meaning two people lie, therefore Liam at the restaurant lied and Tala lied (so Ryan is truthful) therefore so is Isabella. And Charlie who said that Beatriz lied.

9. **Grace and Zara:** Grace at the ice skating rink says Zara at the bowling alley tells the truth. Since Zara lies, Grace also lies.

Therefore: * Grace (ice skating rink) lies. * Zara (bowling alley) lies. * Ryan (vineyard) tells the truth.

**no, no, yes**

---

## Rewritten Candidate Response

**Theorem F.8** (Main Claim). *Given the set of statements made by individuals located at different places, where each individual either always tells the truth or always lies, determine whether Grace (ice skating rink), Zara (bowling alley), and Ryan (vineyard) tell the truth. The answer is: Grace lies, Zara lies, Ryan tells the truth.*

*Proof.* By Lemma F.9, Jaxon lies.
Given that Jaxon lies, by Lemma F.10, Zara lies.
Since Jaxon lies, and Zara lies, by Lemma F.12, Ryan tells the truth.
Since Zara lies, and Grace says Zara tells the truth, by Lemma F.19, Grace lies.
Therefore, Grace lies, Zara lies, and Ryan tells the truth. This corresponds to the answer **no, no, yes**.
□

**Lemma F.9.** *Jaxon lies.*

*Proof.* The problem states "The person at the skate park lies." The problem also states "Jaxon is at the skate park." Therefore, Jaxon lies. □

**Lemma F.10.** *If Jaxon lies, then Zara lies.*

*Proof.* The problem states "The person at the bowling alley thinks their neighbor is telling the truth." Zara is at the bowling alley. Jaxon is Zara's neighbor at the skate park. If Jaxon lies, then Zara must believe a lie, implying Zara lies. □

**Lemma F.11.** *If Zara lies, and Zara says Hiroshi tells the truth, then Hiroshi lies.*

*Proof.* Zara lies, and Zara says Hiroshi tells the truth. Since Zara lies about Hiroshi telling the truth, Hiroshi must lie. □

**Lemma F.12.** *If Zara lies, and Zara says Ryan lies, then Ryan tells the truth.*

*Proof.* Zara lies, and Zara says Ryan lies. Since Zara lies about Ryan lying, Ryan must tell the truth. □

**Lemma F.13.** *If Ryan tells the truth, and Ryan says Tala lies, then Tala lies.*

*Proof.* Ryan tells the truth, and Ryan says Tala lies. Since Ryan tells the truth about Tala lying, Tala must lie. □

**Lemma F.14.** *If Tala lies, and Tala says Priya tells the truth, then Priya lies.*

*Proof.* Tala lies and says Priya tells the truth. Since Tala lies about Priya telling the truth, Priya must lie. □

**Lemma F.15.** *If Priya lies, and Priya says Max lies, then Max tells the truth.*

*Proof.* Priya lies and says Max lies. Since Priya lies about Max lying, Max must tell the truth. □

**Lemma F.16.** *If Max tells the truth, and Max says Elowen tells the truth, then Elowen tells the truth.*

*Proof.* Max tells the truth, and Max says Elowen tells the truth. Since Max tells the truth about Elowen telling the truth, Elowen tells the truth. □

**Lemma F.17.** *If Elowen tells the truth, and Elowen says Charlie tells the truth, then Charlie tells the truth.*

*Proof.* Elowen tells the truth and says Charlie tells the truth. Since Elowen tells the truth about Charlie telling the truth, Charlie tells the truth. □

**Lemma F.18.** *If Charlie tells the truth, and Charlie says Beatriz lies, then Beatriz lies.*

*Proof.* Charlie tells the truth and says Beatriz lies. Since Charlie tells the truth about Beatriz lying, Beatriz lies. □

**Lemma F.19.** *If Zara lies, and Grace says Zara tells the truth, then Grace lies.*

*Proof.* Zara lies, and Grace says Zara tells the truth. Since Grace claims the liar Zara tells the truth, Grace lies. □

