# OpenReview forum: "Sample, Scrutinize and Scale: Effective Inference-Time Search by Scaling Verification"
_ICML.cc/2025/Conference — ICML 2025 poster_

### Official Review · Reviewer_uCGZ · 2025-03-02

**Overall Recommendation:** 3

**Summary:**

The paper studies a minimalist implementation of test-time scaling that uses only random sampling and direct self-verification. The contributions include:
1. the paper shows that the sample-verification method is surprisingly effective, and it is beneficial to scale both number of solution samples per question and number of verification samples per solution.
2. the paper proposes two effective methods to improve self-verification: pairwise response comparison and response style rewriting.
3. the paper introduces a benchmark to evaluate the verification accuracy of LLMs.

## update after rebuttal
I agree with the clarification of the main contributions. Therefore I raised my score to 3. However, I still think the implicit scaling is not reasonable and the dataset lacks diversity.

**Claims And Evidence:**

The paper claims the existence of implicit scaling of verification accuracy: increasing the number k of generated responses increases the conditional probability of at least one of the correct responses is of the highest verification score given that at least one response is correct (Pass@k=1). This is not intuitively or theoretically verified and only supported by the limited evidence on a single model. I note that the trend on the MATH benchmark in Figure 2 is downwards when k is larger than 80, contradicting the claim. Also, the figure omits questions that Consistency@200 answers correctly to reduce noise. It might also change the results.

**Essential References Not Discussed:**

The difference with Snell et al., 2024. is not sufficient discussed.

**Experimental Designs Or Analyses:**

Yes. The experimental setting is simple and solid.

**Methods And Evaluation Criteria:**

The idea of scaling number of verification samples is sensible to improve the verification accuracy. The two methods to improve self-verification also make sense.

The benchmarks used to evaluate reasoning ability are AIME, MATH, and LiveBench. They are common benchmarks in mathematical reasoning. Including reasoning benchmarks in other areas like LiveCodeBench (for coding reasoning) and GPQA (for science reasoning) will be better.

**Other Comments Or Suggestions:**

The authors should place section 4 (Technical Details) in the appendix and appendix A (Related Work) in the main paper.

**Other Strengths And Weaknesses:**

I think the paper is a mixture of contributions to three rather independent directions, while the contribution to each direction is insufficient.

First, the paper highlights the impact of scaling verification compute besides inference compute. However, it is obvious that increasing verification samples leads to higher accuracy, since verification is also a reasoning task and majority voting improves its accuracy. Since there are two scaling factors (number of solution samples per question and number of verification samples per solution) and they both require computing resources, the key problem is what is the optimal allocation between the two factors given a limited compute budget.

Second, the paper proposes two effective methods to improve self-verification. However, their contribution to the end-to-end accuracy is not measured. Also, the pairwise comparison is only applied to the tie-break stage for three highest scoring candidates.

Third, the paper introduces a benchmark to evaluate the verification ability of LLMs. The verification candidates in the benchmark are all sampled from Gemini-1.5-pro, which means that it cannot comprehensively evaluate verification accuracy on diverse solutions.

**Questions For Authors:**

How is the threshold 0.05 selected in the algorithm?

**Relation To Broader Scientific Literature:**

The effectiveness of search against a verifier is already demonstrated in Snell et al., 2024. They also show that search against a verifier can improve accuracy when majority voting saturates.

The verification benchmark is in fact a benchmark for reward model on mathematical reasoning. It is known that the accuracy of reward model is sensitive to the distribution of responses (Zheng et al., 2024). The benchmark consists of only responses from Gemini-1.5-pro. I think it is insufficient to evaluate frontier models.

Snell, C., Lee, J., Xu, K., and Kumar, A. Scaling llm test-time compute optimally can be more effective than scaling model parameters, 2024.
Zheng, C., Zhang, Z., Zhang, B., Lin, R., Lu, K., Yu, B., Liu, D., Zhou, J. and Lin, J., 2024. Processbench: Identifying process errors in mathematical reasoning. arXiv preprint arXiv:2412.06559.

**Theoretical Claims:**

No proofs in the paper.

---

> ### Author Rebuttal · Authors · 2025-04-01
>
> Thank you for your review.
>
> > The main contribution of the paper.
>
> We view our main contribution as showing that scaling search with self-verification works on frontier models and providing an explanation for why: implicit scaling. While it may seem obvious that “increasing verification samples leads to higher accuracy”, many papers (including Snell et al.) have studied self-consistency and self-verification and attempted to scale them up, but none have reported the same success. In fact, we aren’t aware of any prior works that have successfully applied self-verification to frontier models in a way that actually beats self-consistency. Instead, to beat self-consistency, the field has turned to training process-based reward models, using RL to learn verifiers, and other complicated interventions. “Increasing verification samples” sounds like a simple panacea but making it work is a different matter.
>
> This has led to a general belief that simple self-verification isn’t enough, whereas we show that it is—you just need to hit a certain threshold of scale and obey some important principles. It's also noteworthy that direct self-verification can successfully and reliably extract o1-level performance from a non-reasoning model without any finetuning, distillation, RL, or custom-trained verifiers (again, no other works/methods—verification-based or otherwise—have achieved this).
>
> We therefore respectfully, but strongly disagree with the assessment that “the key problem is what is the optimal allocation between the two factors given a limited compute budget”; prior to this work, it wasn’t even known that near-unbounded compute budget is sufficient to recover o1-level performance without interventions like RL.
>
> > “The difference with Snell et al., 2024.”
>
> Our paper differs from Snell et al. in several important ways. First, we study entirely different questions: Snell et al. asks what the optimal test-time scaling trade-off is between pretraining versus inference-time scaling; we ask what the scaling trends and limits of sampling with self-verification are. Second, we study qualitatively different regimes: making verification improve on self-consistency is easy when trying to improve e.g. MATH performance from 10-40% (as is done in Snell et al. 2024); making verification improve on self-consistency on the last 90% of MATH or on actually challenging benchmarks like AIME is significantly more difficult (we aren’t aware of any works prior to ours that were able to do so without RL). Third, Snell et al. 2024 trains a process-based verifier reward model; we show that direct self-verification works fine.
>
> > “The benchmark consists of only responses from Gemini-1.5-pro. I think it is insufficient to evaluate frontier models.”
>
> Gemini 1.5 Pro-002 was released in September 2024, making it one of the main frontier models at the time of this submission. Our paper also includes results on Gemini-1.5-Flash (a significantly smaller model with major qualitative differences) in Appendix B.1. We also note for context that the Snell et al. 2024 paper ran experiments only on the Palm-2 model and on a single benchmark (MATH).
>
> > Omit questions that Consistency@200 answers correctly to reduce noise. It might also change the results.
>
> On MATH for example, 98% of “ambiguous questions” are easy questions that Consistency@200 answers correctly and hence Verification@200 answers correctly. When we plot Figure 2 without omitting these questions, we end up with a line that goes from 98% to 99% which entirely misses the point of the plot (which is to highlight implicit scaling). To fulfill the reviewer’s request, we will include an analog of Figure 2 in the Appendix where Consistency@200 filtering is not done. However, no trends change. For example, on AIME, Consistency@200 only gets 1-2/15 questions correct—omitting those questions has no material impact on the trends observed in Figure 2.
>
> > MATH trend on Figure 2.
>
> Increasing the number of generated samples makes verification on ambiguous questions  harder; this follows from basic probability: if you allow students more attempts at an exam, the average pass-rate of a student who passes at least one exam decreases. Similarly, as you increase the number of attempts, the average number of correct attempts/question decreases. Thus, if implicit scaling didn’t exist, *all* lines in Figure 2 should be strictly decreasing. The fact that we observe the opposite proves that implicit scaling must be happening. The fact that there is a 63->61% drop on the right-hand side of the MATH plot is not evidence against this and is—if anything—expected: we can’t expect implicit scaling is universally powerful enough to completely reverse the natural bias of increasing verification difficulty.

---

> > ### Comment · Reviewer_uCGZ · 2025-04-02
> >
> > About the benchmark, what I mean is that as a benchmark for verification it should cover diverse solutions, not limited to solutions from a single model. It is OK as empirical results to support the idea in the paper, but not enough as a standalone benchmark.
> >
> > I think the paper doesn't give a clear definition of implicit scaling. In the abstract implicit scaling is defined as "sampling a larger pool of responses in turn improves verification accuracy". How is the verification accuracy defined? It is pairwise accuracy given two randomly sampled response? Also, on line 80, the paper claims that implicit scaling is because well-written responses are easier to verify than poorly written responses, and scaling sampling widens the pool of well-written candidates. However, scaling sampling also widens the pool of poorly written responses. If you consider pairwise accuracy, it should be the same.

---

> > > ### Author Response · Authors · 2025-04-05
> > >
> > > **Benchmark.**
> > > Thanks for the clarification and apologies for misunderstanding. We agree that's a reasonable point to make; for example, verification performance on reasoning-model (o1/r1/gemini thinking) solutions may differ from those of base models. Our motivation for providing the benchmark was mainly to highlight some of the failure modes we encountered, and hence focused on 1.5 solutions. We are augmenting the dataset per your suggestions: we have already curated an additional set of accompanying entries for the Gemini 2.0-Flash Thinking model, and are currently procuring R1 scores as well.
> > >
> > > **Implicit Scaling.**
> > > We can define implicit scaling as follows:
> > > * Let $Q_k$ denote the questions on which at least one of the model's $k$ solutions are correct.
> > > * Let $x_k$ denote the fraction of questions in $Q_k$ where the verifier correctly identifies a correct solution from the $k$ candidates (verification accuracy).
> > > * Implicit scaling is where $x_k$ increases with $k$ in expectation.
> > >
> > > It's commonly believed that $x_k$ (verification accuracy) should decrease with $k$; this is provably the case if verifiers select randomly; you are indeed correct that this is also the case if you define verification accuracy as pairwise comparison accuracy.
> > > However, we show $x_k$ actually increases with $k$, even though larger $k$ does increase the pool of poorly written responses.
> > > Note that $x_k$ is what we truly care about with verification---we do use pairwise comparisons, but only as a means of improving $x_k$.
> > > We appreciate the feedback on clarifying "verification accuracy" and will add an extended form of this discussion to our text.

---

### Official Review · Reviewer_CzVL · 2025-03-09

**Overall Recommendation:** 3

**Summary:**

This paper studies the inference-time scaling of LLMs for reasoning tasks in a sampling-based search setting. The authors first study the test-time scaling along two important dimensions, search (number of sampled candidates) and verification (number of verification scores computed). While scaling in both axes improves the accuracy, the authors additionally discovered an implicit scaling where larger sampling pools actually increase the verification accuracy. Built upon their findings, this paper identified two principles to boost self-verification of LLMs, candidate pair comparison and response rewriting.

**Claims And Evidence:**

1. I am confused about the claim "Frontier language models demonstrate a remarkable mismatch between the problem-solving capabilities and poor out-of-box verification capabilities". I am wondering how the "poor out-of-box verification capabilities" are proved in the paper using experiments.

**Essential References Not Discussed:**

The literature discussion/related work is thorough in my opinion.

**Experimental Designs Or Analyses:**

The experiment design in this paper is valid and reasonable. I have some additional suggestions for experiments in the following weaknesses/question section.

**Methods And Evaluation Criteria:**

1. The authors proposed Verification@k as a variant in sampling-based search to study self-verification in LLMs, which is sound and interesting.
2. The proposed two principles are well supported by experiments on the newly constructed benchmark.
3. Benchmark selection for MATH, AIME, and LiveBench is reasonable.

**Other Comments Or Suggestions:**

1. Algorithm 1 may need some improvement to make it easier to understand for readers. For example, it took me some time to understand that "1[issicorrect?]" is actually a binary classification of the given query for verification.
2. This may sound a bit demanding, but I think the paper would be further benefited from adding analyses of R1/R1-Zero models. For example, the distilled version of R1, and analyze their behaviours w.r.t. sampling-based search.
3. Since the authors mention that "rewriting responses with increasing rigour and structure", I am not sure if the current implementation/design in Table 4 actually reflect this.

**Other Strengths And Weaknesses:**

1. I appreciate the author's reporting compute/token consumptions in Section 4. Given the expensive cost of the proposed verification scaling, I would suggest the authors to also report the token consumption of baselines (e.g., Consistency@k) as well for a more fair comparison.
2. The paper offers interesting insights for researchers and practitioners that may be beneficial for downstream tasks. For example, the insights on Pass@k v.s. Pass@1.
3. The paper is overall well-written and is a joy to read.

**Questions For Authors:**

See the above reviews.

**Relation To Broader Scientific Literature:**

This paper provides another comprehensive investigation in the area of test-time computing, specifically scaling at inference time.

One of the key contributions is the exploration of scaling with "verification". The authors propose a very intuitive "verification@k" algorithm, which could serve as baseline for follow-up works.

This paper also offers insights that are not seen in previous works, and would be valuable to the community, such as the conclusion in "Pass@k v.s. Pass@1".

The proposed verification dataset and the two design principles are also new and interesting.

**Theoretical Claims:**

This paper does not contain theoretical claims and corresponding proofs.

---

> ### Author Rebuttal · Authors · 2025-04-01
>
> Thank you for your review! We address your questions/comments below.
>
> > “I would suggest the authors to also report the token consumption of baselines (e.g., Consistency@k) as well for a more fair comparison”.
>
> We appreciate the suggestion and agree it would be useful. As our focus was identifying broader scaling trends rather than computational efficiency, we chose to study a minimalist implementation of sampling-based search that leaves significant room for optimization e.g. via prompt caching, delegated search, small verifiers, etc.—all exciting directions for future work as the reviewer has accurately noted.
>
> > Algorithm 1
>
> We appreciate the feedback and will revise the algorithm description accordingly. The notation 1[is correct] would indeed benefit from clarification.
>
> > Adding analyses of R1/R1-Zero models.
>
> This is a great suggestion and something we hope to do. Unfortunately, since R1 was released only 3 days before the ICML deadline, we were unable to include an analysis in this submission but agree that it’s valuable to analyze R1’s sampling-based search trends.
>
> > On Table 4
>
> Table 4 depicts the implications of ablating the rewriting step from the verification process, showing a significant drop. We include in Appendix 3 a few examples of the outcomes of the rewriting step, which illustrate the rigor and structure that result. We will add a more detailed discussion to this effect as well.

---

### Official Review · Reviewer_VNHP · 2025-03-13

**Overall Recommendation:** 2

**Summary:**

Overall Evaluation
This paper investigates the scalability of sampling-based search methods in inference tasks and proposes a minimal yet effective Sampling-based Search with Self-Verification approach. The key contributions of this work include:

1. A systematic analysis of inference performance scaling with computation, revealing that both search scale (`k_inf`) and verification capacity (`k_verif`) jointly influence reasoning accuracy. The experiments demonstrate that verification plays a crucial role in improving inference reliability.
2. The introduction of the implicit scaling phenomenon (Implicit Scaling), showing that increasing the search scale unexpectedly improves verification accuracy—i.e., as the search space expands, the probability of generating high-quality answers increases, leading to overall performance gains.
3. An evaluation of the out-of-the-box self-verification capability of leading models, showing that GPT-4o and Claude 3.5 Sonnet perform only marginally better than random guessing in self-verification tasks, indicating a significant limitation in current LLM reasoning capabilities.

The paper conducts large-scale experiments on Gemini v1.5 Pro, demonstrating the advantages of Verification@200 over Consistency@200 in complex reasoning tasks. Furthermore, it proposes Comparison-based Verification and Response Rewriting as optimization strategies to enhance model verification.

Overall, the paper is well-structured, with a clear motivation, rigorous experimental design, and insightful conclusions.

**Claims And Evidence:**

Yes

**Essential References Not Discussed:**

NO

**Experimental Designs Or Analyses:**

Yes

**Methods And Evaluation Criteria:**

Yes

**Other Comments Or Suggestions:**

No

**Other Strengths And Weaknesses:**

Comments and Suggestions
- Limited application of Verification@200 on other LLMs: maybe due to constraints in computational resources, the authors did not apply Verification@200 to models such as GPT-4o, which would have provided a broader validation of the method's generalizability. It would be beneficial to explore its impact on multiple state-of-the-art models.
- Positioning relative to alternative reasoning enhancement methods: While Verification@200 demonstrates performance gains, I am concerned about its computational cost compared to alternative reasoning frameworks such as Tree-of-Thoughts (ToT) and reinforcement learning-based approaches. Given the increasing focus on efficient inference, should Verification@200, which requires large-scale computation, be the primary direction for improving model reasoning? A comparative discussion with these alternative methods would strengthen the paper's positioning.

Overall, the work presents a compelling approach to enhancing reasoning through scalable verification but would benefit from additional discussion on computational efficiency and comparisons with alternative reasoning paradigms.

**Questions For Authors:**

See the Comments and Suggestions

**Relation To Broader Scientific Literature:**

Enhance the trustworthy of  LLMs

**Theoretical Claims:**

N/A

---

> ### Author Rebuttal · Authors · 2025-04-01
>
> Thank you for your review! We address your questions below.
>
> > Computational cost compared to alternative reasoning frameworks such as Tree-of-Thoughts (ToT) and reinforcement learning-based approaches.
>
> Optimizing computational efficiency was not the main focus of this paper, which focuses on understanding the broader scaling trends underlying sampling-based search via self-verification. While we chose to study a minimalist implementation of sampling-based search, we expect our insights to generalize to alternative implementations that place a premium on computational efficiency. In fact, there are several avenues along which the effectiveness and efficiency of our minimalist sampling-based search implementation can be improved: prompt caching, delegated search, training smaller verifiers, etc.—all exciting directions for future work. Regarding ToT and RL, these are complementary techniques: you can always apply sampling-based search and self-verification to RL-trained models that use ToT for inference. As a result, comparing between two methods that can be used in concert imposes an unnecessary trade-off.
>
> > Application of Verification@200 on other LLMs.
>
> In Appendix B.2, we include results that apply Verification@200 to the significantly smaller Gemini 1.5 Flash model. Computational concerns were a limiting factor in the number of LLMs studied, but we find no reason for the trends we observed with 1.5 Pro and 1.5 Flash to not generalize to models like GPT4o.

---

### Official Review · Reviewer_VnJ3 · 2025-03-13

**Overall Recommendation:** 3

**Summary:**

This paper examines scaling test-time compute through a sampling and self-verification approach (“Verification@k"). The authors demonstrate that with sufficient sampling and self-verification, even standard models (Gemini v1.5 Pro) can outperform specialized reasoning models (o1-Preview), and Verification@k improves naturally with increased sampling --- whereas Consistency@k plateaus. The authors also propose a verification benchmark showing current large language models (LLMs) exhibit weak out-of-the-box verification abilities.

**Claims And Evidence:**

I’m skeptical of the author’s implicit scaling claim— that increasing the number of generated responses increases the quality of responses. It is unclear if increased generations improves generation quality or simply increases generation coverage [1], and thus even if you have a suboptimal verifier, it is more likely to select a solution that reaches a correct final solution. I don’t follow the authors’ claim that increasing generations will increase the verifier’s likelihood of error, given that pass@k is known to increase– thus, instead of focusing on ‘ambiguous’ questions in Figure 2, I believe it is more interesting to identify the number of instances where the correct solution appears very few (1-2 times) in the k_inf generations, but the verifier still selects the correct answer (the authors provide a singular example of this in Table 2, but there lacks further analysis on this).

[1] Brown, Bradley, et al. "Large language monkeys: Scaling inference compute with repeated sampling." arXiv preprint arXiv:2407.21787 (2024).

Edit: After discussions with the authors, I believe that while the results are not very surprising (ie. you need model scale and proper output formatting for verification to scale with inference generations) I am more sympathetic to the authors demonstrating that this is possible. I still believe the paper requires restructuring for clarity and more rigorous definitions of eg. implicit scaling, but authors have said they will implement these changes.

**Essential References Not Discussed:**

Authors have not cited related work [1] which was previously mentioned above.

[1] Brown, Bradley, et al. "Large language monkeys: Scaling inference compute with repeated sampling." arXiv preprint arXiv:2407.21787 (2024).

**Experimental Designs Or Analyses:**

See my comment in “Claims” regarding the experiments around ‘implicit scaling’. I believe the analysis regarding the accuracy rates relative to scaling number of generations and number of verifications could be improved with a better FLOPs comparison to judge the best allocation of verification and generation attempts; the authors mention compute for a fixed setting of the parameters on AIME in section 4, but this should be present across their results in Section 2.

**Methods And Evaluation Criteria:**

Evaluation criteria is straightforward (correctness on the reasoning benchmarks considered). I'm seeking clarification for one part of the methodology: the authors propose a “Verification@k” process detailed in Algorithm 1, but upon reading the details in Section 3.1 the authors state that the candidate responses are rewritten in a format with Theorems/Lemmas/Proofs before being passed for self-verification. This is not obvious from the introduction, and this confounds the verification process with an additional step that may be aiding self-verification. There are not many details about how the rewrites are prompted, even though the authors do ablate over variations of this prompt.

**Other Comments Or Suggestions:**

No other comments, see below for questions.

**Other Strengths And Weaknesses:**

Careful studies of how search should be scaled at inference time and whether self-verification can be effectively leveraged is a relevant area of research. The authors present analysis on several standard math and reasoning datasets. The potential of self-verification for models to reach reasoning-model performance is promising.

The observation that reasoning models have weak out-of-box verification capabilities has been previously reported (eg [1]). Further, I believe the paper is organized poorly to the point that it is confusing for a reader to follow the paper. The introduction presents a result (Table 1) before the methods are described, and only Algorithm 1 is given before the authors dive into more results in Section 2, only to loop back to talking about the ‘Sampling-Based Search Implementation’ in subsection 3.1 and technical details in Section 4. Further, the search method employed in Section 2 appears to not be the same as that in Algorithm 1; the tiebreaking was omitted, which I only understood in full after reading Line 275 in Section 3.2. I ask the authors to reorganize their paper such that the methods and setup are described before results are presented.

Given my feedback about the exposition, the claims, and the insufficient analysis, I do not recommend acceptance of the paper in its current state. I am willing to raise my score if several points raised in my review are clarified, if the paper’s structure is reorganized, and some additional analyses are provided (see Questions).

[1] Wu, Zhenyu, et al. "Large language models can self-correct with key condition verification." arXiv preprint arXiv:2405.14092 (2024).

**Questions For Authors:**

1. Are the results in Section 2 doing the same process as in Section 3.1? In Step 2, it is mentioned that the scoring attempt is a rewrite of the response as a theorem, lemmas and proofs. How are these rewrites done? If this was done for Section 2, this should be explicitly stated because it is not clear from Algorithm 1 that this is what is done, and  this confounds the verification process by presenting it in a standardized format which potentially makes verification easier.
2. Can the authors clarify their point about implicit scaling and how their results suggest higher quality of generations with multiple samples?
3. Do you have a FLOPs comparison of the different settings of k_inf and k_ver? It personally is not surprising to me that scaling up both k_inf and k_ver will show increases in performance, and the analysis feels incomplete without showing a FLOPs tradeoff between inference and verification generations to show, given a fixed compute budget, what the tradeoff should be between generation and verification.
4. On each benchmark, for the questions in which the number of solutions which reached the correct are very low (eg. 1-2 generations), what fraction does Verification@k get correct?
5. Could the authors clarify what is prompted and given to the model during self-verification for the 'main' prompt and each of the three variations? Why are the FP and FN rates missing for the LiveBench questions? Does the difference in FP and FN rates across the different prompts translate to significant differences in the Verification@k accuracy?

**Relation To Broader Scientific Literature:**

There is a growing trend of using additional computation at inference to enhance performance. The paper directly compares to prior scaling test-time methods like Consistency@k. There has also been recent attention in AI research aimed at making LLMs more trustworthy and less prone to errors. Self-verification is a prominent approach under exploration.

**Theoretical Claims:**

N/A, there are no theoretical claims.

---

> ### Author Rebuttal · Authors · 2025-04-01
>
> Thank you for your review. We address your questions below.
>
> > “I’m skeptical of the author’s implicit scaling claim… it is unclear if increased generations improves generation quality or simply increases generation coverage.”
>
> We understand this concern and provided Figure 2 for exactly this reason. Figure 2 explicitly controls for the fact that coverage (Pass@k) increases with k by limiting our evaluation to questions on which the model has at least *one* correct answer (i.e., “ambiguous questions”). In fact, we also provided Figure 4 for this reason, which is exactly the same as Figure 1 except that it controls for “generation coverage”. Re Brown et al.: Brown et al. is concerned with the scaling of Pass@k (usually an unrealistic upper bound and not an operative metric); we are concerned with how much of the Pass@k - Pass@1 gap can actually be attained.
>
> > “I don’t follow the authors’ claim that increasing generations will increase the verifier’s likelihood of error, given that pass@k is known to increase.”
>
> The only problems on which verification can err are ambiguous problems (where there is at least one correct answer) and increasing k can make verification on ambiguous questions  harder. This latter claim follows from basic probability: if you allow students more attempts at an exam, the average pass-rate of a student who passes at least one exam decreases. Similarly, as you increase k, the average number of correct solutions/question decreases. Thus, if implicit scaling didn’t exist, the lines in Figure 2 should all provably be non-increasing or even decreasing.
>
> > “The observation that reasoning models have weak out-of-box verification capabilities has been previously reported”
>
> We agree and note in the paper that “the limitations of model self-verification capabilities are well-studied”. This is why we believe our results are so surprising: while prior works have attempted to scale self-verification on frontier models, none have succeeded in beating self-consistency with direct self-verification and instead needed to turn to masking techniques, learning verifiers, process-based rewards, etc. In contrast, we show that direct self-verification can be scaled to reliably achieve o1-level performance without finetuning, RL, distillation, or custom models—something that our paper is the first to do. While we already cite 30+ papers on LLM self-verification, we will add a brief discussion on Wu et al. which is largely orthogonal to this work.
>
> > Paper structure.
>
> We appreciate the feedback and will look at revising the structure of the paper per your advice. Since the algorithm we present is fairly minimal, we wanted to highlight the scaling trends first.
>
> —
>
> Questions:
>
> > Are the results in Section 2 doing the same process as in Section 3.1?
>
> Section 2 omits tie-breaking for computational reasons but is otherwise the exact same process as Section 3.1 (including rewriting). Regarding rewriting, we see it as just part of the verification process—one that is sadly often overlooked. We don’t see how it undermines/confounds any of our reported results, e.g. implicit scaling. We provide the rewriting process in the source code (it’s pretty generic; we use the exact same instructions for all benchmarks), but will revise to include a copy in the Appendix.
>
> > Do you have a FLOPs comparison of the different settings of k_inf and k_ver?
>
> We are unable to publicly report the FLOPs used in closed-source models, but do detail the number of output tokens used (from which FLOPs can be extrapolated) and will include a more involved discussion. We focused on closed-source models in this paper as—at the time of this submission—no OSS models had reached the minimum level of capability needed for self-verification to scale.
>
> > For the questions in which the number of solutions which reached the correct are very low (eg. 1-2 generations), what fraction does Verification@k get correct?
>
> We will run a more formal analysis, but we can give some preliminary numbers. In particular,  the AIME exam falls in the regime you’re asking about: on 12/15 questions, less than 4% of Gemini Pro solutions are correct. Nonetheless, Verification@200 gets >50% accuracy. On the 7/15 questions where exactly 1-4% of questions are correct, Verification@200 gets 5/7 correct. We agree that this would be impactful to include and appreciate the suggestion.
>
> > Could the authors clarify what is prompted during self-verification...? Why are the FP and FN rates missing for the LiveBench questions? Does the difference in FP and FN rates translate to... Verification@k accuracy?
>
> The full prompts for all four options referenced in Table 4 can be found in the source code; we will revise to include a copy in the Appendix. We added LiveBench at a later date and had not yet re-run Table 4, but will include them in our revision. We informally observed Verification@k accuracy to be especially sensitive to FN rates, and will add a formal table to this effect.

---

> > ### Comment · Reviewer_VnJ3 · 2025-04-05
> >
> > Thank you to the authors for their response. I am more sympathetic to the authors clarifying the main contributions of their paper, which show that self-verification can be successful with sufficient scale and certain prompt rewritings.
> >
> > * I believe part of the confusion for the results, methods and claims of the paper is due to the structure. I understand that the algorithm is not incredibly complex but requires more clarification and context given that other reviewers have also brought this up. Along with properly explaining the algorithm before diving into results, I also believe the authors need to highlight the rewriting mechanism more because it seems this was an important part of the process (regarding Table 4 results and the fact it is 'often overlooked')
> > * When the authors say "no OSS models had reached the minimum level of capability needed for self-verification to scale", what is this 'minimum level of capability'? It seems that for verification@k accuracy to increase with k, one needs a model with sufficient self-verification capabilities as well as a fat enough tail over outputs such that one is more often sampling a correct, high-quality response.
> >
> > I am willing to increase my score if the authors agree to revise the structure of the paper for further clarity and include the missing results.

---

> > > ### Author Response · Authors · 2025-04-05
> > >
> > > We appreciate the detailed feedback around the paper's structure and have indeed been revising the structure accordingly. It's actually quite refreshing to have such direct, helpful and actionable feedback.
> > >
> > > Thank you for the opportunity to clarify:
> > > even if we have a set of candidate solutions containing good solutions (e.g. generated by a capable enough model with a fat enough tail), we need a verifier model that meets a "minimum level of capability". Specifically, if the verifier model is sufficiently poor at verification, it may end up behaving like random sampling---producing results actually inferior to self-consistency---or worse (for example, weak models will often bias towards long, detailed solutions where the generating model actually went off the rails). It's hard to formally quantify this "level of capability", but we consider MATH and LiveBench Reasoning performance to be a reasonable proxy and anecdotally estimate GPT-4o/Gemini 2.0-Flash to be just above this minimum level.

---

### Official Review · Reviewer_LFT3 · 2025-03-13

**Overall Recommendation:** 3

**Summary:**

This paper claims that while self-consistency can greatly improve LLM performance, leveraging additional test time compute to verify/compare generated responses can break the plateau for self-consistency and further enhance model performance, the paper conducts extensive experiments to validate their findings and propose a verification benchmark to evaluate the models’ verification capabilities.

**Claims And Evidence:**

Yes, the claims made in the submission are supported by clear and convincing evidence.

**Essential References Not Discussed:**

This paper discussed the related works detailedly.

**Experimental Designs Or Analyses:**

Yes, I have reviewed the validity of the experimental designs and find them well-constructed and sufficient to support the paper's claims.

**Methods And Evaluation Criteria:**

Yes, proposed methods and/or evaluation criteria make sense.

**Other Comments Or Suggestions:**

If there are any misunderstandings on my part, please point them out, and I will reconsider my evaluation of this work.

**Other Strengths And Weaknesses:**

**Strengths**

1. This paper identifies several previously unstudied principles of self-verification, including implicit scaling, output style suitability, and the importance of directly comparing responses, which is interesting and worth investigating, and promising to enhance LLM reasoning ability.
2. This paper provides a verification benchmark to evaluate LLM self-verification abilities.

**Weaknesses**

While the authors explicitly state that their work focuses on demonstrating scaling trends in LLM self-verification rather than optimizing specific strategies, I have two main concerns:

1. Self-verification appears to be a subset of LLM self-consistency, making the observed scaling trends somewhat predictable.
2. Although the authors differentiate their work through discoveries about output style suitability and the importance of direct response comparison, these contributions may not be substantial enough.

**Questions For Authors:**

I have to questions:

1.  The relatively small gap between Pass@1 and Consistency@5 in Table 5 seems to contradict the paper's main findings about scaling trends. Could the authors explain this?
2. In Table 5, the random guessing entries show an 80%/20% split for Correct/Wrong responses rather than the expected 50%/50%. Could the authors explain this more detailedly?
3. Is there the evaluation result for Gemini v1.5 in Verification Benchmark?

**Relation To Broader Scientific Literature:**

The finding that models' verification abilities can be enhanced through increased response sampling has potential for improving model performance.

**Theoretical Claims:**

The paper focuses on empirical finding, no theoretical proofs or claims.

---

> ### Author Rebuttal · Authors · 2025-04-01
>
> Thank you for your review.
>
> To address your comments on “predictability”:
> Many papers have studied self-consistency and self-verification, and attempted to scale them up. None have reported the same success that we have; in fact, we aren’t aware of any prior works that have successfully applied self-verification to frontier models in a way that actually beats self-consistency. Instead, to beat self-consistency, the field has turned to training process-based reward models, using RL to learn verifiers, and other complicated interventions.
>
> This has led to a general belief that direct self-verification isn’t enough, whereas we show that it is—you just need to hit a certain threshold of scale and obey some important principles. While this may be intuitive, it’s not at all “predictable”. It's also noteworthy that direct self-verification can successfully and reliably extract o1-level performance from a non-reasoning model without any finetuning, distillation, RL, or custom-trained verifiers (again, no other works/methods—verification-based or otherwise—have achieved this).
>
> To your second concern:
> As we note above, our main contribution is showing that scaling search with self-verification *does work* on frontier models and providing an explanation for why: implicit scaling. Methods like direct comparisons are ways of taking advantage of implicit scaling, and are a big reason why our attempts at scaling self-verification were uniquely successful.
> For example, it is notoriously difficult to beat self-consistency when it comes to the MATH performance of frontier models that already get 90%+ accuracy (hence our paper is the first to do so). As we show in Table 3, direct comparison plays a key role in overcoming this “last-mile” barrier, enabling self-verification to actually beat self-consistency.
>
> —
>
> Individual questions:
>
> > “The relatively small gap between Pass@1 and Consistency@5 in Table 5.”
>
> The small gap between Pass@1 and Consistency@5 indicates that scaling self-verification accuracy via self-consistency is limited in effectiveness—this doesn’t contradict any of our findings. In fact, it’s part of our core argument: the techniques we propose (like direct comparisons) allow for more effective test-time scaling of self-verification than self-consistency.
>
> > "In Table 5, the random guessing entries show an 80/20 split for Correct/Wrong responses rather than the expected 50/50. Could the authors explain?"
>
> 50/50 and 80/20 are both attainable with random guessing (just change the bias of the coin you flip). Because the prior of frontier models is skewed towards 80/20, we wanted to provide a similarly calibrated random baseline and so opted to display 80/20 instead of 50/50.
>
> > "Is there the evaluation result for Gemini v1.5 in Verification Benchmark?"
>
> Good question; we exclude Gemini v1.5 from the benchmark since we use Gemini v1.5 to generate the candidate solutions. Evaluating Gemini v1.5 on the benchmark seemed unsound given the potential confounding.

---

### Decision · Program_Chairs · 2025-05-01

**Decision:**

Accept (poster)

**Comment:**

This paper studies test-time compute, and finds that generating lots and lots of candidate solutions combined with self verification allows smaller LLMs to outperform state-of-the-art reasoning models.  The findings here are interesting and topical.  Given the interest in reasoning and test-time compute right now, this paper will contribute to ICML.  The reviewers largely voted for acceptance, with the one dissenting reviewer only presenting two comments, neither of which I feel are disqualifying.  For example, the reviewer suggests that this paper should investigate computational efficiency.  While this would be interesting, it is not the focus of the paper and I think the paper stands without that.  Therefore, I vote for acceptance.